# Neuron-wide RNA transport combines with netrin-mediated local translation to spatially regulate the synaptic proteome

Sangmok Kim[1], Kelsey C Martin[1,2,3]*

[1]Department of Biological Chemistry, University of California, Los Angeles, Los Angeles, United States; [2]Department of Psychiatry and Biobehavioral Sciences, University of California, Los Angeles, Los Angeles, United States; [3]Integrated Center for Learning and Memory, David Geffen School of Medicine, University of California, Los Angeles, Los Angeles, United States

**Abstract** The persistence of experience-dependent changes in brain connectivity requires RNA localization and protein synthesis. Previous studies have demonstrated a role for local translation in altering the structure and function of synapses during synapse formation and experience-dependent synaptic plasticity. In this study, we ask whether in addition to promoting local translation, local stimulation also triggers directed trafficking of RNAs from nucleus to stimulated synapses. Imaging of RNA localization and translation in cultured Aplysia sensory-motor neurons revealed that RNAs were delivered throughout the arbor of the sensory neuron, but that translation was enriched only at sites of synaptic contact and/or synaptic stimulation. Investigation of the mechanisms that trigger local translation revealed a role for calcium-dependent retrograde netrin-1/DCC receptor signaling. Spatially restricting gene expression by regulating local translation rather than by directing the delivery of mRNAs from nucleus to stimulated synapses maximizes the readiness of the entire neuronal arbor to respond to local cues.

*For correspondence:
kcmartin@mednet.ucla.edu

## Introduction

Synapse formation and long-lasting experience-dependent synaptic plasticity require new RNA and protein synthesis (*Kandel, 2001*; *Kalinovsky and Scheiffele, 2004*). Neurons are dramatically polarized and compartmentalized cells, elaborating axonal and dendritic processes that extend long distances and forming thousands of synaptic connections with other neurons. This polarity and compartmentalization poses challenges to the spatial regulation of gene expression during synapse formation and synaptic plasticity: how can the products of gene expression be restricted to specific synapses undergoing activity-dependent changes in structure and function? Many studies have demonstrated that mRNAs that constitutively localize to axonal growth cones in immature neurons and to dendrites in mature neurons undergo stimulus-induced translation, demonstrating that regulated translation of localized mRNAs provides one mechanism of spatially restricting neuronal gene expression (*Aakalu et al., 2001*; *Job and Eberwine, 2001*; *Leung et al., 2006*; *Lyles et al., 2006*; *Smith et al., 2005*; *Tsokas et al., 2005*; *Wang et al., 2009*; *Zheng et al., 2001*). Here, we set out to determine whether, in addition to regulating compartmentalized translation, local stimulation also regulates RNA trafficking from the soma to the synapse. We considered that the regulated trafficking of stimulus-induced transcripts from nucleus to stimulated synapse would provide a means of directly coupling the requirement for transcription with the requirement for local translation during synapse formation and plasticity.

A number of findings indicate that the persistence of long-term plasticity requires not only translation but also transcription. For example, transcriptional inhibitors block late-phase LTP of rodent

**eLife digest** The nervous system contains billions of cells called neurons that connect to each other to form complex networks via junctions known as synapses. Synapses form between the end of an elongated section (called the axon) of one neuron and the tiny projections (or dendrites) from the cell body of the next neuron.

Throughout the lifespan of an animal, the nervous system responds to experiences and stimuli from the environment by changing the strength of the connections at synapses; this is known as 'synaptic plasticity'. In this way, long-term information about learning and memory can be stored and used to direct future responses to similar situations.

Many proteins are involved in forming and altering synapses. The genes that code for these proteins are found in the nucleus of the neuron within the cell body. To make new proteins, copies of genes are made using molecules called mRNAs, which then leave the nucleus and are used as templates by the machinery that assemble proteins. Previous studies have shown that mRNA molecules are transported from the cell body to the axon and dendrites, but it is not clear exactly where the proteins are produced.

Kim et al. have now studied the movement of mRNAs in neurons from the sea slug *Aplysia* during synapse formation and synapse plasticity. This showed that mRNAs are delivered equally throughout the neuron, and so it appears that mRNAs are not targeted to a particular synapse. However, the level of protein production using these mRNA molecules is much higher in places where synapses are being formed or altered. A protein called netrin-1 promotes protein production in the dendrites of neurons at these synapses.

Kim et al. demonstrate that although mRNAs are delivered throughout the neuron, they are only used to make proteins at specific synapses. This allows the entire neuron to be in a state of readiness to make new synapses or alter existing ones in response to stimuli from the environment. Understanding more about how this local control of protein production works within neurons may provide new insights into diseases that affect synaptic plasticity.

hippocampal synapses (*Frey et al., 1996*; *Nguyen et al., 1994*) and long-term facilitation (LTF) of *Aplysia* sensory-motor synapses (*Montarolo et al., 1986*), indicating that translation of pre-existing localized mRNAs is not sufficient for late-LTP in hippocampus or for the persistence of LTF in *Aplysia*. Similarly, when isolated *Aplysia* sensory neurites, severed from their somata, are paired with motor neurons, they undergo a form of serotonin (5HT)-induced LTF that does not last as long as the LTF in cultures containing intact SNs (*Grabham et al., 2005*). Together these studies demonstrate that newly transcribed mRNAs are required for persistent synaptic plasticity. This led us to ask whether, in addition to regulating local translation, synaptic stimulation regulates the trafficking of RNAs from the nucleus to specific subcellular sites.

Consistent with the possibility of stimulus-induced control of RNA trafficking, previous studies have reported that activity can regulate the localization of mRNAs within neurons. For example, depolarization has been shown to increase the dendritic localization of several mRNAs, including those encoding beta-actin (*Tiruchinapalli et al., 2003*), BDNF, and TrkB (*Tongiorgi et al., 1997*), while incubation with DHPG has been shown to increase the dendritic localization of the AMPA glutamate receptor GluA2 (*Grooms et al., 2006*). Similarly, neurotrophins have been reported to regulate the concentration of specific mRNAs in axons of regenerating adult sensory neurons (*Willis et al., 2007*). Neuronal activity also induces transcription of genes whose transcripts are subsequently transported into dendrites, including for example the immediate early gene Arc (*Link et al., 1995*; *Lyford et al., 1995*). In situ hybridization images of the *bclw* transcript in a recent study from *Cosker et al (2013)* indicated that the mRNA localized to peripheral but not central axons of DRG neurons, suggesting that neurons are capable of directing transcripts from the soma to subsets of processes. Together, these findings suggest that RNAs may undergo activity-dependent transport from the nucleus to locally stimulated subcellular compartments.

In this study, we directly monitored RNA targeting from the soma to stimulated synapses during synapse formation and synaptic plasticity. To examine synapse formation, we cultured a single bifurcated *Aplysia* SN with a target (L7) MN, with which it formed glutamatergic synapses, and a non-target

(L11) MN, with which it fasciculated but did not form chemical synapses (*Glanzman et al., 1990*). To study RNA localization during synapse-specific plasticity, we cultured bifurcated SNs with two target MNs and locally perfused 5HT onto the connection with one MN to produce synapse-specific LTF. We asked whether and how stimuli regulate RNA and protein localization and concentration by analyzing the distribution of ribosomal RNA (rRNA), messenger RNA (mRNA), RNA binding proteins, and translation factors. We found that rRNA, three specific mRNAs, ribosomal proteins, and translation factors were delivered throughout the SN, to both synaptic and non-synaptic sites, but that translation was significantly enriched at sites of synaptic contact and at sites of synaptic stimulation. These results indicate that the spatial regulation of gene expression in *Aplysia* sensory-motor neurons is mediated primarily at the level of translation rather than at the level of RNA targeting from the nucleus. These results, together with our previous study (*Wang et al., 2009*) suggested the existence of a trans-synaptic signal that promotes translation of localized mRNAs. In investigating the nature of such a trans-synaptic signal, we uncovered a role for netrin-1/DCC signaling in local translation at synapses. Our findings are thus consistent with neurons delivering transcripts and translational machinery throughout the neuron, but with a synaptically restricted netrin-1-dependent signal triggering localized translation at synapses.

## Results

### Ribosomal RNA targets equally well to SN branches contacting target and non-target MNs

We first asked whether the most abundant RNA, ribosomal RNA (rRNA) underwent directed targeting from neuronal cell bodies to synaptic sites during synapse formation. Towards this end, we performed fluorescent in situ hybridization (FISH) for 18S and 28S rRNA in cultures (3DIV) containing a bifurcated SN contacting a target L7 MN, with which it formed glutamatergic synapses, and a non-target L11 MN, with which it fasciculated but did not form chemical synapses (*Figure 1A* and *Figure 1—figure supplement 1A*). Quantification of 18S (*Figure 1B,C*) and 28S (*Figure 1—figure supplement 1B,C*) RNA in the proximal SN branches did not reveal any significant difference in concentration in the branch contacting the L7 target MN and the branch contacting the L11 non-target MN. We also examined the distribution of ribosomes by expressing the *Aplysia* ribosomal protein S6 tagged with the fluorescent tag dendra2 in bifurcated SNs contacting both target L7 and non-target L11 MNs (*Figure 1D*). This allowed us to detect the ribosomal protein S6 in distal SNs without any signal from the MN (in the rRNA FISH experiments, we focused on proximal SN processes in order to avoid confounding signal from the MNs). As shown in *Figure 1E,F*, S6-dendra2 targeted equally to distal SN branches contacting target or non-target MNs. Together, these data indicate that ribosomes are delivered throughout the SN and are not targeted specifically from the soma to branches receiving synaptogenic signals.

### Sensorin mRNA targets equally well to SN branches contacting target and non-target MNs

As a candidate localized mRNA, we first examined the transcript encoding the SN-specific neuropeptide sensorin, which not only localizes to distal sensory neurites but also concentrates at synapses. Since sensorin is only expressed in SNs, we could visualize sensorin mRNA in proximal and distal neurites without any background signal from MNs. We have previously shown that the neuritic and synaptic localization of sensorin mRNA are mediated by distinct signals (*Meer et al., 2012*; *Wang et al., 2009*) and that sensorin mRNA undergoes localized translation at synapses during 5HT-mediated LTF (*Wang et al. 2009*). FISH for sensorin mRNA in 3DIV cultures revealed that the sensorin transcript localized equally to both proximal (*Figure 2D*) and distal (*Figure 2B,C*) branches of bifurcated sensory neurons contacting L7 or L11 MNs. The pattern of sensorin mRNA in distal branches was distinct, with diffuse distribution of sensorin mRNA in branches contacting non-target MNs and punctate concentrations of sensorin RNA at SN synapses onto target MNs (see *Figure 2—figure supplement 1A* for quantification of coefficient of variation). However, the same amount of RNA was present in both SN branches. Similar results were observed when we performed FISH analysis on 1DIV cultures (*Figure 2—figure supplement 1B,D*). These findings are consistent with a lack of directed targeting of sensorin mRNA from soma to branches receiving synaptogenic signals, although there is local redistribution of sensorin mRNA within the neuronal process at synaptic sites.

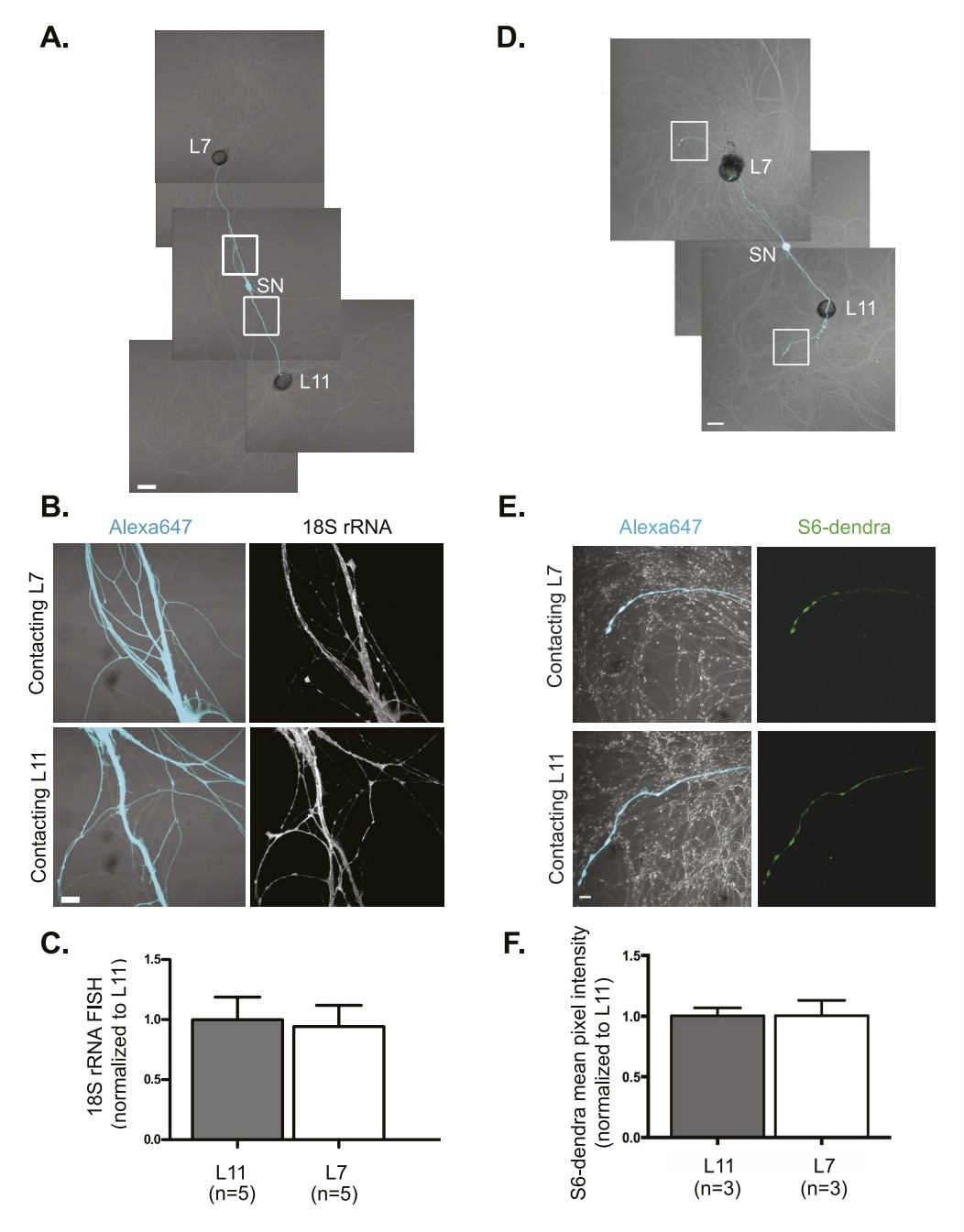

**Figure 1**. rRNA targets equally well to SN branches contacting target and non-target MNs. A bifurcated SN was cultured with an L7 target MN and an L11 non-target MN (**A**) for 3 days. Alexa647 (cyan) was microinjected into the SN before imaging to visualize SN processes. Cultures were processed for Fluorescent *In Situ* Hybridization (FISH) for 18S rRNA (**B**). Areas in proximal SN neurites outlined with white squares in (**A**) were imaged at high magnification in (**B**); left panels show merged images of DIC and Alexa fluorescence, and right panels show FISH signals. The Alexa647 served as a volume control to analyze FISH RNA intensity. Group data show that 18S rRNA (**C**) is evenly distributed in neurites contacting L7 target and L11 non-target MNs. Parallel experiments showing FISH for 28S rRNA are shown in *Figure 1—figure supplement 1*. A bifurcated SN was cultured with an L7 target MN and an L11 non-target MN (**D**) for 1 day and a plasmid expressing ApS6 tagged with dendra2 was microinjected together with Alexa647 (cyan) into the SN. Live imaging (**D**) was performed on day 3; shown in (**E**) are high magnification images of DIC and Alexa fluorescence on the left and ApS6-dendra2 on the right. The Alexa647 served as a volume control to analyze FISH RNA intensity. Group data show that ApS6 (**F**) is evenly distributed in

*Figure 1. Continued on next page*

*Figure 1. Continued*

neurites contacting L7 target and L11 non-target MNs. Error bars represent SEM. None of the differences were significant as determined by a Student's paired *t*-test. Scale bar in (**A** and **D**) =100 μm; in (**B** and **E**) =20 μm.

The following figure supplement is available for figure 1:

**Figure supplement 1**. 28S rRNA targets equally well to SN branches contacting target and non-target MNs.

---

We also examined the distribution of the mRNA encoding β-thymosin, which we had previously identified as a localized mRNA in *Aplysia* SNs (*Moccia et al., 2003*). Since β-thymosin is expressed in both SN and MN, we examined its localization in proximal SN processes. As shown in *Figure 2—figure supplement 2*, β-thymosin mRNA was distributed equally well in SN branches contacting L7 target and L11 non-target MNs.

To assay the localization of an activity-dependent transcript, we performed in situ hybridization for EF1α. EF1α was expressed at very low levels in unstimulated neurons and was distributed equally well between L7 target and L11 non-target branches (*Figure 3A,B*). Its expression was significantly induced following five spaced bath applications of (5HT), as previously described (*Giustetto et al., 2003*). As shown in *Figure 3C,D*, we found that equal amounts of 5HT-induced EF1α were delivered to SN branches contacting L7 target and L11 non-target MNs. These results indicate that activity-induced transcripts are not delivered preferentially from the nucleus towards a synaptic target.

We then asked whether local stimulation might regulate the transport of EF1α from the soma towards stimulated synapses. Towards this end, we cultured a bifurcated SN with two target LFS MNs and used local perfusion to deliver five applications of 5HT to the connections made onto one of the MNs, which we have previously shown produces branch-specific LTF (*Martin et al., 1997*). In control experiments, we locally perfused five applications of artificial seawater (ASW), the vehicle. As shown in *Figure 3*, FISH for EF1α revealed that local stimulation with 5x5HT-induced expression of EF1α, as evidenced in increase in EF1α signal in the soma (*Figure 3G*). Strikingly, the stimulated transcript was delivered equally well to both branches (*Figure 3F,H*). These studies reveal that even following local stimulation, transcriptionally induced mRNAs are delivered throughout the neuronal arbor, to stimulated- and unstimulated-branches.

## RNA binding protein Aplysia Staufen targets equally well to SN branches contacting target and non-target MNs

We next studied the localization of the RNA binding protein Staufen in SNs contacting target and non-target MNs by expressing *Aplysia* Staufen (ApStaufen) tagged with the fluorescent protein dendra2 in bifurcated sensory neurons. We focused on Staufen because of its previously described role in transporting transcripts into dendrites of mammalian cells (*Kiebler et al., 1999*; *Tang et al., 2001*). As shown in *Figure 4*, we found that ApStaufen-dendra2 was transported equally well to branches contacting target and non-target MNs.

We also examined the distribution of ApStaufen-dendra2 in SN paired with two target MNs following local stimulation with 5X5HT to induce branch-specific LTF. As shown in *Figure 4—figure supplement 1*, equal numbers of ApStaufen-dendra2 puncta were present in both SN branches in prior to stimulation and following local perfusion with vehicle (ASW). While local stimulation with 5X5HT increased the number of ApStaufen-dendra2 puncta in the SN processes (suggesting that local stimulation promotes the transport of RNAs out of the soma into the process), the increase was equivalent in stimulated- and unstimulated-SN branches.

## Translation is significantly enriched in SN branches contacting target MNs

Together, our experiments using FISH to analyze rRNAs and sensorin and EF1α mRNAs, and overexpression to analyze S6 ribosomal protein and Staufen, argue that ribosomes, mRNAs, and RNA binding proteins are transported throughout the neuron, without any preferential targeting from the soma to branches receiving synaptic signals. We have previously reported that local translation is spatially restricted to stimulated synapses during synapse-specific long-term facilitation (LTF) of SN-MN synapses (*Wang et al 2009*); here, we asked whether local translation is also spatially restricted during synapse formation. To do this, we performed immunocytochemistry with anti-sensorin antibodies in

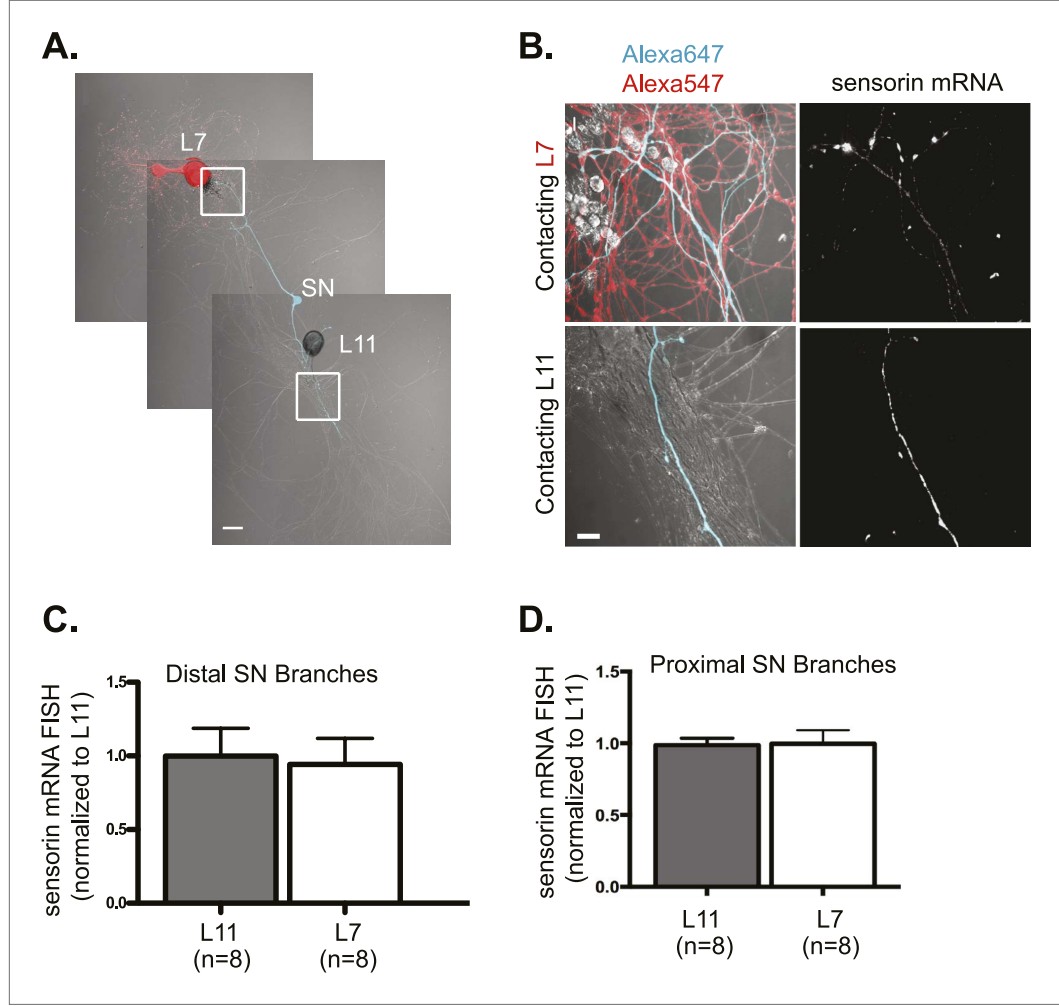

**Figure 2**. Sensorin mRNA targets equally well to SN branches contacting target and non-target MNs. A bifurcated SN was cultured with an L7 target MN and a L11 non-target MN (**A**) for 3 days. Alexa647 (cyan) was microinjected into the SN before imaging to visualize SN processes; in (**A**), Alexa 546 (red) was also microinjected into the L7 MN before imaging to visualize the L7 MN processes. Cultures were processed for Fluorescent In Situ Hybridization (FISH) for sensorin. Areas outlined with white squares in (**A**) was imaged at high-magnification to visualize mRNA in distal (**B**) SN neurites contacting L7 and L11 MNs, respectively; left panels show merged images of DIC and Alexa fluorescence, and right panels show sensorin mRNA FISH signals. The Alexa647 served as a volume control to analyze FISH RNA intensity. Group data show that sensorin mRNA (**C** and **D**) is evenly distributed in proximal and distal SN neurites contacting L7 target and L11 non-target MNs. Error bars represent SEM. None of the differences were significant as determined by a Student's paired *t*-test. The pattern of sensorin mRNA distribution differs in distal branches contacting L7 and L11 MNs, as shown in *Figure 2—figure supplement 1A*, because sensorin mRNA concentrates at synapses (*Lyles et al 2006*). Scale bar in (**A**) =100 µm; in (**B**) =20 µm.

The following figure supplements are available for figure 2:

**Figure supplement 1**. Sensorin mRNA distribution in DIV3 SN-MN cultures and localization in DIV1 SN-MN cultures.

**Figure supplement 2**. β-Thymosin mRNA localizes equally well to SN branches contacting target and non-target MNs.

sensory-motor cocultures (3DIV, *Figure 5*). Again, we chose to focus on sensorin because it is only expressed in the SN and thus the signal derives exclusively from the SN. As shown in *Figure 5B,C*, sensorin protein was present at much higher concentrations in SN branches contacting target MNs

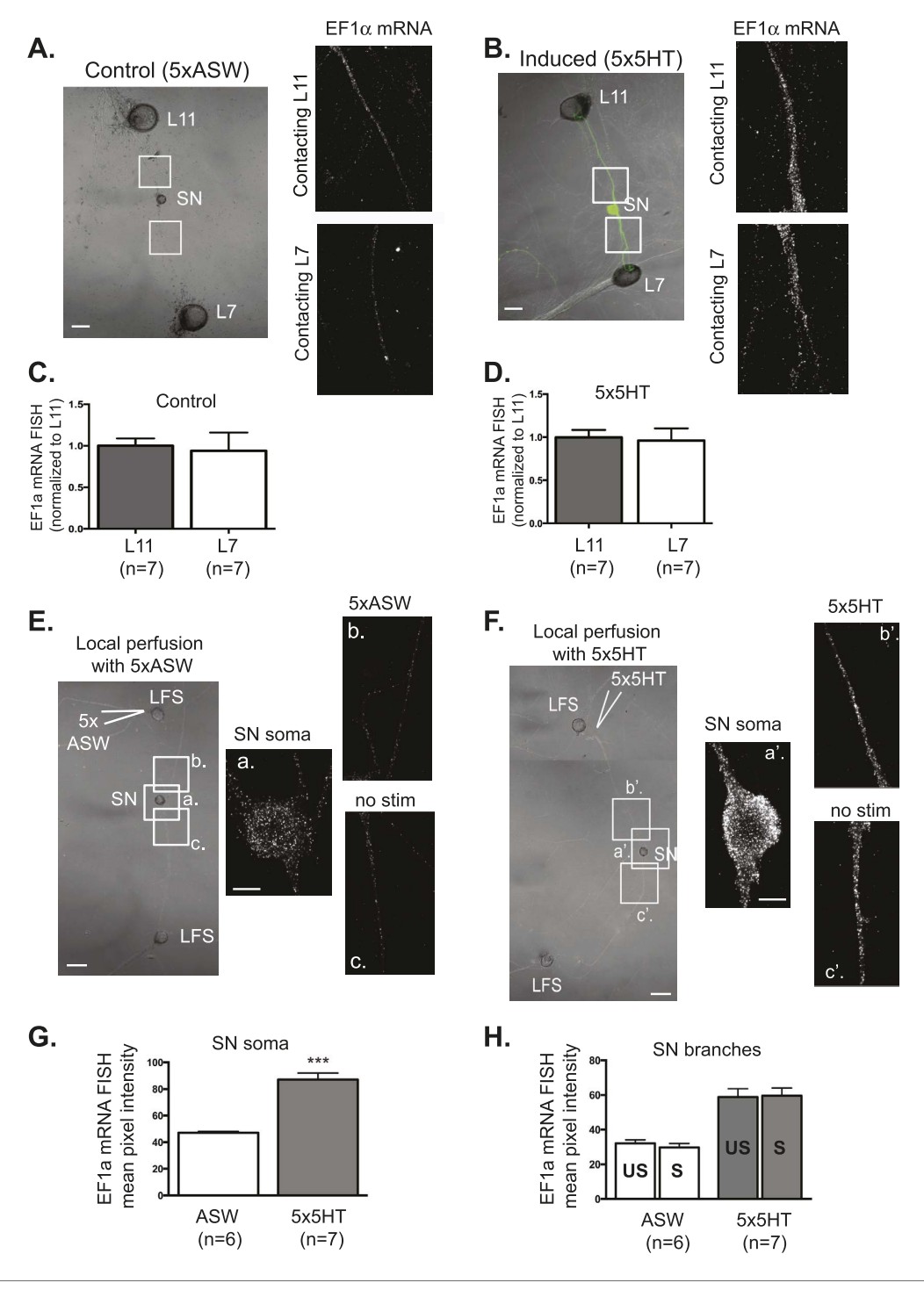

**Figure 3**. EF1α mRNA is induced by 5x5HT and is delivered throughout the neuronal arbor. A bifurcated SN was cultured with an L7 target MN and an L11 non-target MN (**A** and **B**) for 3 days. Alexa647 (cyan) was microinjected into the SN before imaging to visualize SN processes. On day 3, cultures were stimulated with 5 bath applications of 5HT to induce long-term facilitation (LTF) (**B**), or mock stimulated with 5 bath applications of artificial seawater (ASW, the vehicle) as controls (**A**). Four hours later, cultures were processed for Fluorescent In Situ Hybridization (FISH) for EF1α mRNA. Areas in proximal SN neurites outlined with white squares in low magnification DIC image on the left were imaged at high magnification to visualize EF1α mRNA FISH signal. The increase in EF1α mRNA FISH signal in (**B**) as compared to (**A**) indicates that EF1α mRNA was induced by bath application of 5x5HT, as

*Figure 3. Continued on next page*

*Figure 3. Continued*

described in *Giustetto et al (2003)*. Group data show that EF1α mRNA is evenly distributed between SN neurites contacting L7 target and L11 non-target MNs both in control cultures (**C**), and in cultures undergoing 5HT-induced LTF (**D**). A bifurcated SN was cultured with two LFS motor neurons (**E** and **F**) for 3 days. On day 3, local perfusion of five pulses of 5HT was performed to induce branch-specific LTF (**F**), as described in *Martin et al (1997)*; in control cultures, synapses were mock stimulated with five pulses of ASW (**E**). Four hours later, cultures were processed for Fluorescent In Situ Hybridization (FISH) for EF1α mRNA. Areas in soma and proximal SN neurites outlined with lettered white squares in low magnification DIC image on the left were imaged at high magnification to visualize EF1α mRNA FISH signal. The increase in EF1α mRNA FISH signal in (**F**) as compared to (**E**) indicates that EF1α mRNA was induced by local stimulation with 5x5HT. This increase is quantified in SN soma and in proximal neurites in (**G**) and (**H**). As shown in (**H**), basally expressed and local stimulation-induced EF1α mRNA was evenly distributed between 5HT-stimulated (S)- and unstimulated (US)-SN branches. ***p < 0.001, Student's unpaired *t*-test. Differences in FISH signal between mock (ASW)-stimulated and -unstimulated branches (**E**) and 5HT-stimulated and -unstimulated branches (**F**) were not significant, Student's paired *t*-test (**H**). Scale bar in (**A**, **B**, **E** and **F**) =100 μm; in (**E**(**a**) and **F**(**a'**)) =20 μm.

than in branches contacting non-target MNs. These data are consistent with sensorin expression being spatially regulated at the level of translational regulation rather than mRNA targeting.

We previously reported live imaging experiments using a translational reporter consisting of the 5′ and 3′ UTRs of sensorin fused to the photoconvertible fluorescent protein dendra2 to demonstrate synapse-specific translation during synapse-specific LTF of SN-MN synapses (*Wang et al., 2009*). We now conducted live imaging experiments with this reporter to directly measure local translation (as opposed to somatic translation followed by protein transport into the neurite during synapse formation) (*Figure 5D,F*). To do this, we expressed the reporter in bifurcated SNs contacting L7 and L11 MNs and removed the SN cell body so that we could specifically monitor translation in neuronal processes. We photoconverted dendra2 in the branches from green to red and used live imaging to detect the appearance of newly translated, green dendra2 signal, which we have previously demonstrated represents new translation (*Wang et al., 2009*). As shown in *Figure 5E,G*, we detected a significantly greater increase in green signal in SN processes contacting the L7 target MN than in the SN processes contacting the L11 non-target MN. Together, these experiments indicate that although sensorin mRNA is delivered throughout the SN, it is preferentially translated at sites of synaptic contact.

To determine whether this effect was specific to sensorin mRNA, or whether it applied more generally to global translation, we analyzed the localization of the translation eukaryotic initiation factor4E (eIF4E) and the 4E binding protein (4EBP) in bifurcated sensory neurons contacting L7 and L11 motor neurons (for demonstration of antibody specificity, see *Figure 5—figure supplement 1J*). As shown in *Figure 5B,C* and *Figure 5—figure supplement 1A,I*, both eIF4E and 4EBP proteins distributed equally in SN branches contacting target and non-target MNs. However, when we used phospho-specific antibodies that recognize the activated forms of eIF4E (peIF4E) and 4EBP (p4EBP), we detected significantly higher concentrations in SN branches making synaptic contact with L7 MNs. These findings indicate that translational machinery is distributed throughout the neuron but is preferentially activated at sites receiving a synaptogenic signal. Since phosphorylation of eIF4E and 4EBP activates global translation, our results indicate that translational regulation, rather than mRNA targeting, controls the locally regulated synaptic proteome.

As an additional way of measuring translation, we used the recently developed ribopuromycilation method in which immunofluorescence is used to measure puromycin immobilized on ribosomes by the elongation inhibitor emetine. This metabolic labeling method allows detection of newly translated proteins. As shown in *Figure 6*, these experiments revealed significantly more protein synthesis in SN varicosities contacting the synaptic L7 target MN than in SN contacts with the non-target L11 MN. These results provide further evidence that synapse formation does not alter RNA targeting within the neuron but rather spatially regulates translation.

## Netrin-1 promotes translation in SN branches contacting target and non-target MNs

Our previous study using translational reporters to visualize local translation of sensorin mRNA during LTF of *Aplysia* SN-MN synapses (*Wang et al., 2009*) revealed that serotonin-regulated translation

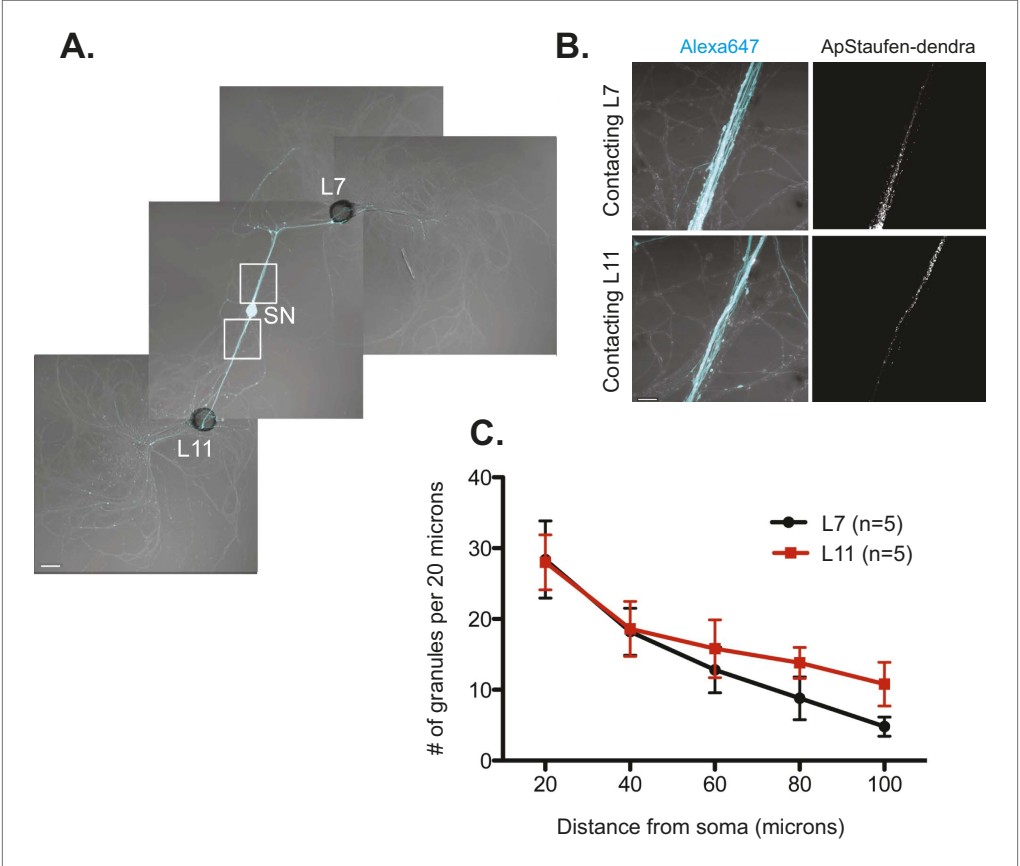

**Figure 4**. The RNA binding protein *Aplysia* Staufen targets equally well to SN neurites contacting L7 target and L11 non-target MNs. (**A**) C-terminally tagged dendra2-tagged *Aplysia* Staufen (ApStaufen) was microinjected together with Alexafluor 647 (cyan) into a cultured bifurcated SN contacting an L7 target MN and an L11 non-target MN at 1DIV. Staufen granules in SN neurites were imaged 48 hr later. Representative high magnification images of areas denoted by white squares in (**A**) are shown in (**B**). Left panels show merged DIC/Alexa fluor images, and right panels show ApStaufen-dendra2 granules images. (**C**) Group data reveal that the number of ApStaufen-dendra2 granules per 20 μm was the same in neurites contacting target L7 MNs (black, closed circle) and non-target L11 MNs (red, square), Student's paired *t*-test at each time point. Error bars represent SEM. Scale bar in (**A**) =100 μm; in B =20 μm.

The following figure supplement is available for figure 4:

**Figure supplement 1**. The RNA binding protein *Aplysia* Staufen targets equally well to stimulated- and unstimulated-SN processes after local perfusion.

specifically at stimulated synapses and that this regulation required a calcium-dependent trans-synaptic signal from the MN to the SN (***Wang et al., 2009***). Our current finding that sensorin protein was enriched at synaptic sites as compared to non-synaptic sites (***Figure 5B,C***) is also consistent with a role for a synaptically localized translational regulation mechanism. These findings suggested that a synaptically generated signal functioned to regulate translation.

We considered the possibility that the guidance factor netrin-1 might serve as such a signal. Netrin-1 is a chemotropic factor known to induce synaptogenesis in *Caenorhabditis elegans* (***Colón-Ramos et al., 2007***) and to stimulate translation in growth cones during axon guidance (***Campbell and Holt, 2001***). Flanagan and colleagues have reported that components of the translational machinery, including ribosomal subunits and translation factors, are tethered at the plasma membrane in neuronal dendrites by binding to the cytoplasmic tail of the netrin receptor Deleted in Colorectal Cancer (DCC), (***Tcherkezian et al., 2010***). Netrin-1 binding was shown to trigger the release of bound translational components and to thereby promote localized, netrin-1- dependent protein synthesis.

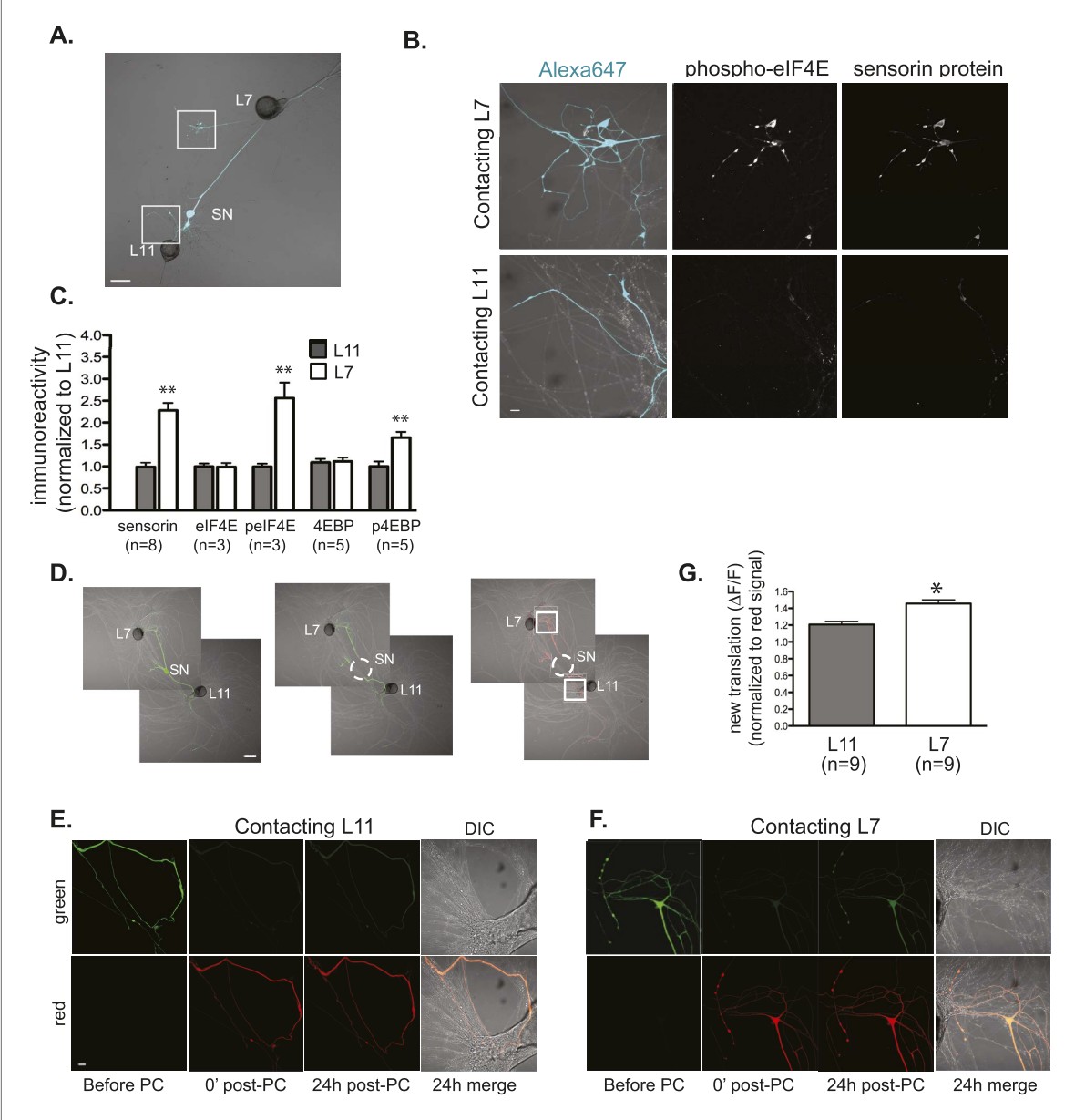

**Figure 5**. Local translation is greater in SN neurites contacting L7 target MNs than in neurites contacting L11 non-target MNs. (**A**) A bifurcated SN was cultured with an L7 target MN and an L11 non-target MN for 3 days. Alexa647 was microinjected into the SN and served as a volume filling control for analysis of immunocytochemistry (ICC). To monitor local translation, we performed ICC for phosphorylated eIF4E (phospho-eIF4E), phosphorylated 4EBP, and sensorin protein. Representative high magnification images of areas denoted by white squares in (**A**) are shown in (**B**), with merged DIC/Alexa fluor in the left panel, phospho-eIF4E ICC in the middle panel, and sensorin ICC in the right panel. Group data in (**C**) reveal that phospho-eIF4E, phospho-4EBP, and sensorin protein concentrations are significantly greater in neurites contacting L7 target MNs than in neurites contacting L11 non-target MNs. As shown in *Figure 5—figure supplement 1*, total eIF4E and total 4EBP protein were distributed equally well in neurites contacting L7 target and L11 non-target MNs. To directly image local translation, we expressed a translational reporter consisting of sensorin mRNA fused to the photoconvertible fluorescent protein dendra2 (as described in *Wang et al. (2009)*), removed the SN soma (dotted circle in middle panel of **D**), and 12–18 hr later photoconverted the dendra2 signal from green to red (right panel of **D**). Shown in (E, neurite contacting L11) and (F, neurite contacting L7) are high magnification images of regions marked by white squares in (**D**), including images before photoconversion (pre), right after PC (post), 24 hr later (24 hr post). The top panels show the green channel and the bottom show the red channel (which was used as volume control for quantification of the green signal). Increased green signal represents newly translated reporter. Group data (**G**) show that there is significantly more translation, measured as ΔF/F, in neurites contacting L7 target MNs than in neurites contacting L11 non-target MNs. Error bars represent SEM. **$p < 0.05$, unpaired t-test. Scale bars in (**A**) and (**E**) =100 μm; in (**C**), (**D**), and (**F**) =10 μm.

*Figure 5. Continued on next page*

*Figure 5. Continued*

The following figure supplement is available for figure 5:

**Figure supplement 1**. Local translation is enriched in SN branches contacting L7 MN.

Together, these findings suggested that release of netrin-1 from either SN or MN during synapse formation and during synaptic plasticity triggers translation at sites of synaptic but not non-synaptic contacts.

We tested the possibility that netrin-1 regulates the synthesis of localized transcripts by incubating *Aplysia* sensory-motor neuronal cultures with recombinant Fc-netrin-1. As shown in *Figure 7A,C*, bath application of netrin-1 (250 ng/ml) for 24 hr increased sensorin immunoreactivity in SN branches contacting the L11 non-target MN, such that there was no longer any branch-specificity of sensorin protein concentration.

To further examine the effect of netrin-1 on sensorin expression, we cultured SNs with target MNs in the presence or absence of netrin-1 and compared sensorin immunoreactivity. As shown in *Figure 7D,E*, incubation with netrin-1 significantly increased sensorin immunoreactivity. To differentiate between stimulated local translation and stimulated somatic translation followed by transport into the branches, we cultured bifurcated SN-MN, removed the SN cell body, incubated with netrin-1, and processed the cultures 24 hr later for sensorin immunoreactivity (*Figure 7—figure supplement 1A,C*). These experiments revealed that netrin-1 increased sensorin concentration equally well in branches contacting both L7 and L11 MNs, consistent with local stimulation of translation. Netrin-1 also increased peIF4E immunoreactivity in SN-LFS cultures in which the SN soma had been removed (*Figure 7—figure supplement 1C*). The netrin-1-induced increase in sensorin immunoreactivity in intact SN-LFS cocultures was similar to that observed with five spaced applications of serotonin (5HT), which produces LTF of SN-MN synapses and which has been shown to promote sensorin translation (*Figure 7E*) (*Hu et al., 2006*; *Wang et al., 2009*). Moreover, netrin-1-induced translation in intact or soma-lacking SNs was blocked by pre-incubation with the protein synthesis inhibitor anisomycin (10 µM for 24 hr), indicating that the increased immunoreactivity did indeed result from increased translation (*Figure 7E* and *Figure 7—figure supplement 1C*).

We considered the possibility that netrin-1 might transform non-synaptic sites to synapses and that the increase in translation occurred as a result of the conversion of non-synaptic to synaptic sites. However, incubation of SN-L11 cocultures or L7-SN-L11 cocultures with netrin-1 did not result in the formation of synapse as shown by the absence of an excitatory post-synaptic potential (EPSP) in the L11 MN after SN stimulation (data not shown). We did find, however, that incubation with netrin-1 significantly increased EPSP amplitude between SN and LFS target MNs (*Figure 7F,G*). The increase in EPSP amplitude was similar to that seen during LTF induced by five spaced applications of 5HT (data not shown). LTF induced by five applications of 5HT has also been shown to involve significant increases in the number of SN varicosities (sites of synaptic connections) in SN-MN cultures (*Glanzman, et al., 1990*). We thus measured the effect of netrin-1 on varicosity numbers, and, as shown in *Figure 7H*, found that it significantly increased the number of SN synaptic varicosities in SN-LFS MN cultures. Together, these findings suggest that the local translation induced by netrin-1 increases synaptic strength and synapse number.

## Netrin-1 binds to pre-synaptic DCC receptors to promote translation

To explore the mechanisms by which netrin-1 regulates translation, we first asked whether it did so by binding to DCC. To test this, we incubated cultured neurons with recombinant netrin-1 in the presence of monoclonal anti-DCC antibodies that have previously been shown to block netrin-1 binding to the DCC receptor (*Bennett et al., 1997*; *Braisted et al., 2000*; *Manitt et al., 2009*). We first showed that a 24 hr incubation of isolated SNs with netrin-1 triggered a significant increase in sensorin immunoreactivity, but that incubation of isolated SNs with anti-DCC (in the absence of netrin-1) did not change sensorin immunoreactivity (*Figure 8A,B*). While anti-DCC antibodies had no effect on the basal concentration of sensorin in isolated SNs, they significantly decreased basal sensorin immunoreactivity in SN-MN cocultures (*Figure 8C,D*). Taken together, these findings are consistent with netrin-1 being endogenously released from the post-synaptic MN and binding to DCC on the SN to promote pre-synaptic translation of sensorin.

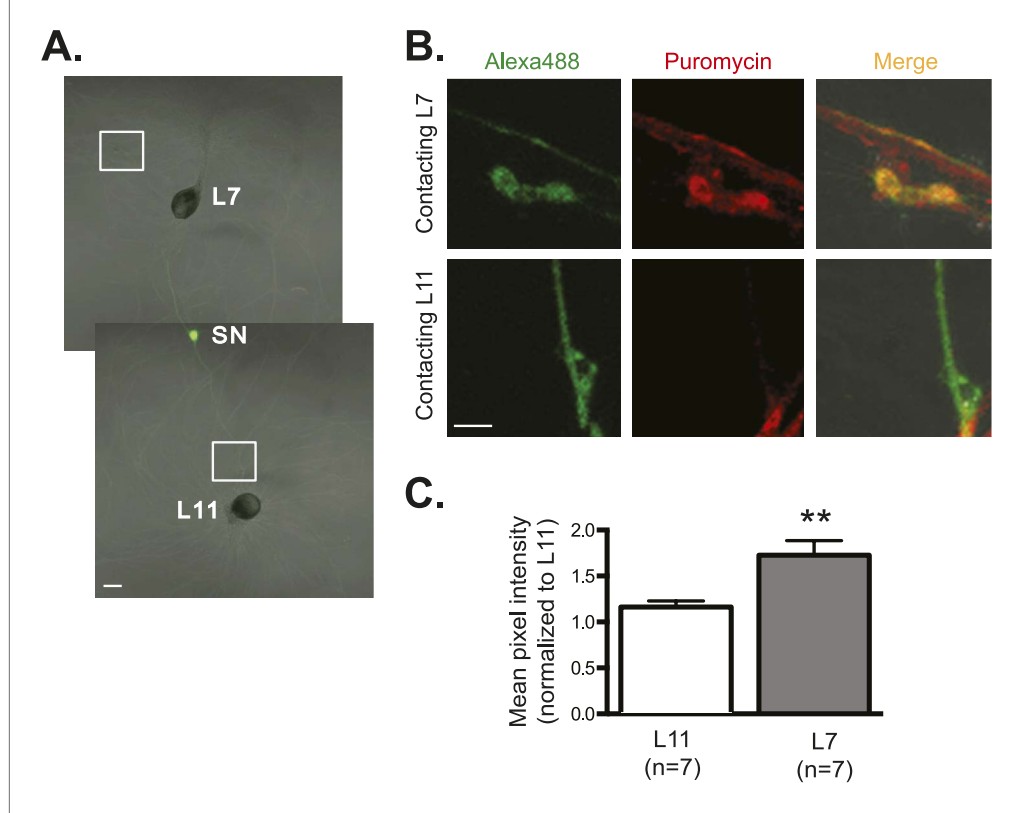

**Figure 6**. Global translation is also enriched in SN branches contacting L7 MN. A bifurcated SN was cultured with an L7 target MN and an L11 non-target MN (**A**) for 3 days. Alexa488 (green) was microinjected into the SN as a volume control for imaging (**A**). Cultured neurons were processed for ICC for puromycin. 100 µM puromycin was applied to cultured neurons for 10 min in the presence of 200 µM emetine. Puromycin incorporated into newly synthesized peptide was detected by anti-puromycin antibody. Representative high magnification images of areas denoted by white squares in (**A**) are shown in (**B**), with Alexa488 (green) in the left panel, anti-puromycin in the middle panel (red), and merged images in the right panel. Arrows indicated puromycin immunoreactivity in SN varicosity (**B**) in the middle panel. Group data (**C**) show that the puromycin incorporation is greater in SN branches contacting L7 target MNs than in branches contacting non-target L11 target MNs. Error bars represent SEM. *p < 0.05, paired *t*-test. Scale bars in (**A**) =100 µm; in (**B**) =10 µm.

We next explored the function of netrin-1/DCC signaling during synapse formation. Towards this end, we cultured a SN for 24 hr, then added a MN in the presence or absence of vehicle (ASW), control IgG (non-immune mouse IgG), or function-blocking anti-DCC antibodies, and measured EPSP amplitude 24 hr later. As shown in *Figure 8H*, EPSP amplitude was significantly lower in cultures incubated with anti-DCC antibodies than in control cultures. This was accompanied by a decrease in sensorin immunoreactivity in the cultures incubated with anti-DCC antibodies (*Figure 8C,D*). To determine whether netrin-1/DCC also regulated synapse maintenance and/or stability, we cultured SN-MN neurons for 2 days, added anti-DCC antibodies, and then measured EPSP amplitude 24 hr later. As shown in *Figure 8E*, this also triggered a significant decrease in EPSP amplitude (*Figure 8H*) and sensorin immunoreactivity (*Figure 8D*), consistent with netrin-1/DCC signaling contributing to synapse stability/maintenance.

To further examine netrin-1 and DCC expression in Aplysia SN-MN cocultures, we cloned the Aplysia homologs (submitted to GenBank). Aplysia Netrin-1 (ApNetrin-1, accession #KM218335) was 47% identical and 60% conserved with human Netrin-1, and Aplysia DCC (ApDCC, accession # KM218336) was 31% identical and 47% conserved with human DCC. To assay the effect of ApNetrin-1 on synaptic strength, we overexpressed c-terminally tagged dendra2-ApNetrin-1 in MNs (with overexpression of dendra2 as a negative control) and determined the effect on sensorin immunoreactivity (as a proxy for translation) and SN-MN strength. As shown in *Figure 9*, overexpression of

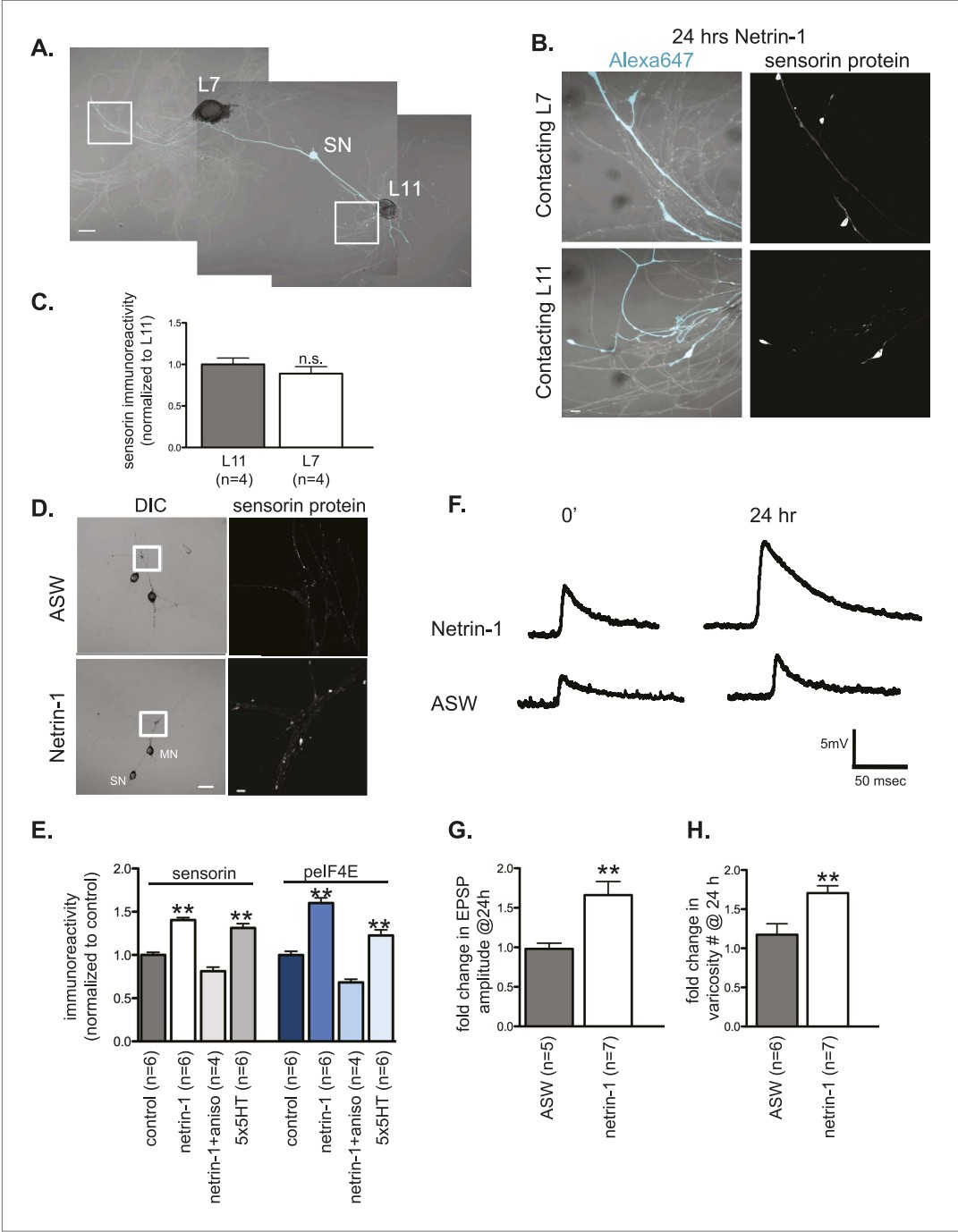

**Figure 7**. Netrin-1 increases protein synthesis in SNs and strength of SN-target MN synapses. (**A**) A bifurcated sensory neuron was cultured with an L7 target MN and an L11 non-target MN for 3 days. Alexa647 was microinjected into SNs and served as a volume filling control for the analysis of immunocytochemistry (ICC). Cultures were incubated with 250 ng/ml recombinant Fc human netrin-1 for 24 hr. Representative high magnification images of areas marked by white squares in (**A**) are shown in (**B**), with merged DIC/Alexa488 in the left panels, and sensorin ICC in the right panels. Group data (**C**) reveal that the concentration of sensorin in netrin-1-treated SNs is the same in neurites contacting L7 target MNs and L11 non-target MNs. We compared sensorin (**D**) and phospho-eIF4E immunoreactivity (not shown) in SN-LFS target MN cultures in the presence and absence of 250 ng/ml Fc-netrin-1 (24 hr). Representative high magnification images of areas denoted by white squares in left panels of (**D**) stained with anti-sensorin antibodies are shown in right panels of (**D**). Group data (**E**) reveal that netrin-1 triggers an increase in sensorin and phospho-eIF4E immunoreactivity that is equivalent to the increase observed 24 hr after 5 spaced

*Figure 7. Continued on next page*

*Figure 7. Continued*

applications of 5HT (which produce long-term facilitation) and that is blocked by the protein synthesis inhibitor anisomycin (10 mM). The effect of netrin-1 on sensorin immunoreactivity in isolated SN processes and on eIF4E and phospho-eIF4e immunoreactivity is shown in *Figure 7—figure supplement 1*. Netrin-1 also increased synaptic strength: representative traces of EPSPs evoked in LFS MNs after stimulation of SN, at time 0 and 24 hr after incubation with vehicle (artificial seawater, ASW) or netrin-1 are shown in (**F**); histogram of group data is shown in (**G**). Histogram in (**H**) shows fold change in the number of varicosities between SN and LFS MN after 24 hr of incubation with vehicle (ASW) and netrin-1. Error bars represent SEM. **p < 0.001, Student's unpaired *t*-test for (**G**) and (**H**); ANOVA and Dunnett's multiple comparison test for (**E**). Scale bars in (**A**) and (**D**, left panel) =100 μm; in (**B**), (**D**, right panel)= 10 μm.

The following figure supplement is available for figure 7:

**Figure supplement 1**. Netrin-1 increases local translation in soma-free SN processes in SN-MN cultures.

dendra2-tagged ApNetrin-1 in MNs generated a punctate staining pattern throughout the neuronal process. Overexpression of ApNetrin-1 in the MN significantly increased the concentration of sensorin protein in the SN as measured by immunocytochemistry (*Figure 9B,C*), consistent with a netrin-1-dependent stimulation of translation. Measurement of EPSP amplitude before and 24 hr after overexpression of dendra2-ApNetrin-1, or overexpression of dendra2 as a negative control, revealed that ApNetrin-1 significantly increased synaptic strength between SN and MN (*Figure 9E*).

Since our previous study had indicated that local translation required a calcium-dependent retrograde signal from the MN (*Wang et al., 2009*), we next asked whether ApNetrin-1 release from the MN was a calcium-dependent process. To do this, we expressed ApNetrin-1 tagged with dendra2 in the MN, and labeled the MN with Alexa fluor 647 and the SN with ApDCC tagged with mCherry. As shown in *Figure 10*, despite being expressed in the MN, the majority of the ApNetrin-1-dendra2 signal was found outside of the MN, decorating the processes of the SN, suggesting that ApNetrin-1 was released from the MN and bound to the SN. We note that overexpression of ApDCC-mCherry in the SN increased the ApNetrin-1 signal that decorated the SN (as compared to the signal that was observed in SNs labeled with red Alexa647, as in *Figure 9*), and thus allowed us to clearly detect binding of green ApNetrin-1 expressed in the MN to the red SN. To determine whether ApNetrin-1 bound to DCC on the SN, we performed these experiments in cultures containing anti-DCC antibodies in the medium (or non-immune IgG as a negative control). As shown in *Figure 10—figure supplement 1*, incubation with anti-DCC antibodies dramatically decreased the binding of ApNetrin-1 to the SN, as assayed using Pearson's Correlation to monitor the colocalization of green ApNetrin-1-dendra2 and red DCC-mCherry.

To test whether release from the MN required calcium, we microinjected 50 mM BAPTA into the MN and 3 hr later imaged green ApNetrin-1-dendra2 signal localization. As shown in *Figure 10B,D*, when calcium was chelated in the MN, green ApNetrin-1-dendra2 signal was sequestered in the MN and did not colocalize with the SN. In contrast, in the absence of BAPTA, the green ApNetrin-1-dendra2 signal decorated the processes of the SN. Together, these findings are consistent with calcium-dependent release of ApNetrin-1 from the MN and binding to the SN.

## Discussion

This study was aimed at determining whether localized stimulation, during synapse formation or synaptic plasticity, triggered directed mRNA targeting out of the nucleus and whether such directed RNA trafficking contributed to the spatial regulation of neuronal gene expression. Using a simple system consisting of a single, bifurcated *Aplysia* SN contacting a target L7 MN, with which it formed glutamatergic synapses, and a non-target L11 MN, with which it did not form chemical synapses, we asked whether and how synaptogenic signals regulate RNA localization and/or local translation. To determine how local stimulation spatially regulates gene expression, we cultured a single bifurcated SN with two target MNs and locally perfused 5HT onto the SN connections that formed onto one MN to produce synapse-specific LTF (*Martin et al., 1997*). We used 18S and 28S rRNA FISH and S6-dendra2 overexpression to monitor the localization of rRNA; sensorin, β-thymosin and EF1α FISH to monitor localization of three mRNAs, including one transcriptionally induced mRNA; and Staufen-dendra2 overexpression to monitor localization of an RBP previously reported to be involved in mRNA localization

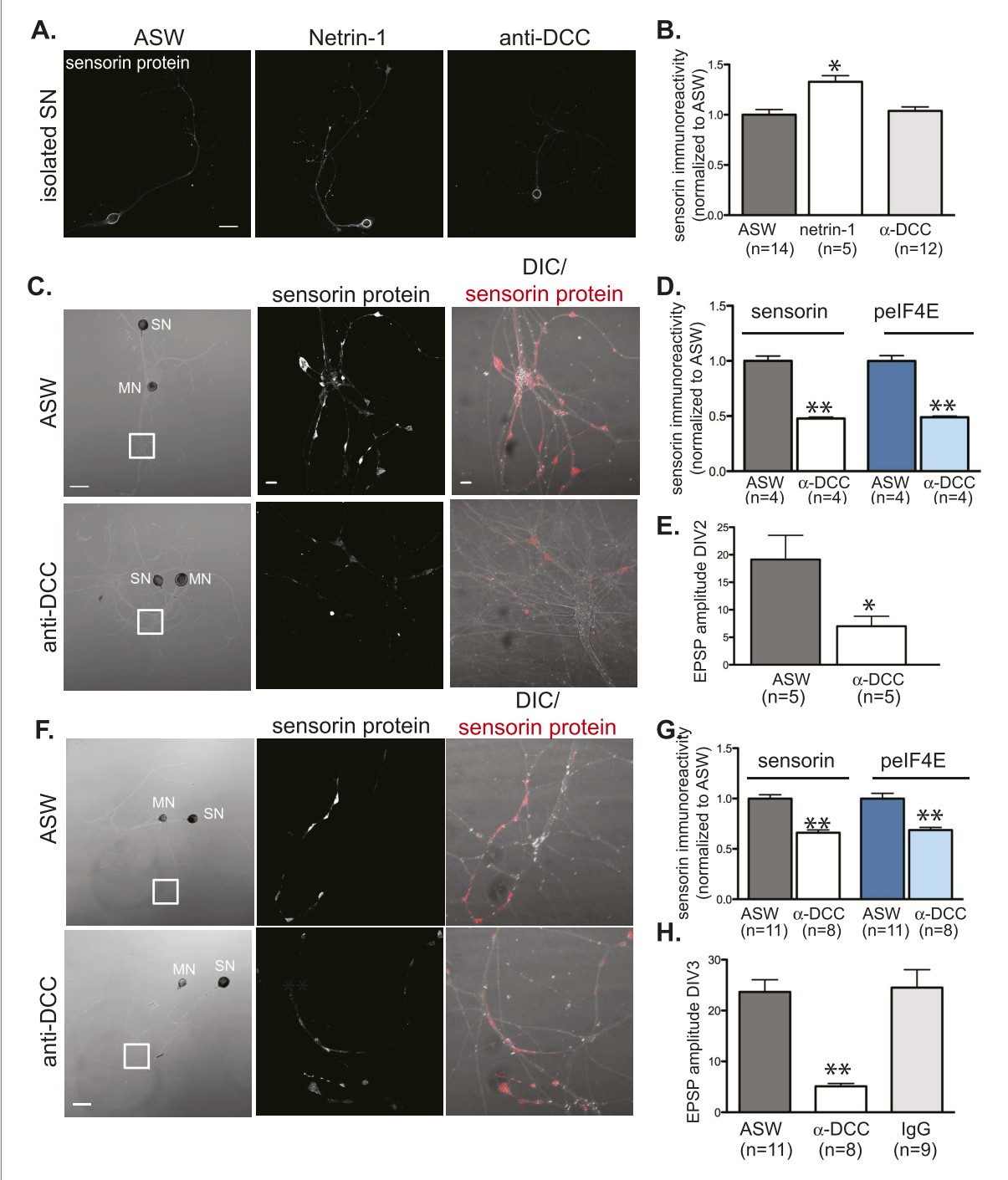

**Figure 8**. Netrin-1 binds DCC to promote SN translation and to increase strength of SN-MN synapses. (**A**) and (**B**): immunocytochemistry (ICC) of isolated SNs (3DIV) reveals an increase in sensorin immunoreactivity with recombinant Fc human netrin-1 incubation (250 ng/ml for 24 hr), but no change from baseline following 24 hr of incubation with function-blocking anti-DCC antibodies (250 ng/ml). Representative images are shown in (**A**); group data in (**B**). To test the effect of function-blocking anti-DCC antibodies on sensorin and phospho-eIF4E (peIF4E) immunoreactivity and synapse formation in SN-LFS target MN cocultures, we cultured isolated SNs for a day, added anti-DCC (250 ng/ml), and 6 hr later paired the SN with a target LFS MN. EPSP amplitude was recorded the next day, and cultures were fixed for ICC. Representative cultures are shown in (**C**). High magnification images of areas denoted by white squares in left DIC images are shown in the middle (sensorin ICC) and the right (merged DIC and sensorin, red) panels. Group data (**D**) show that anti-DCC significantly decreases sensorin and peIF4E immunoreactivity, consistent with netrin being released from the MN to drive translation in the SN. SN-LFS MN pairs incubated with anti-DCC antibodies also showed significantly smaller EPSP amplitudes than do control cultures (**E**). To query

*Figure 8. Continued on next page*

*Figure 8. Continued*

the function of netrin-1/DCC signaling in mature, established SN-LFS MN synapses, we cultured SN-LFS MNs for 2 days, added anti-DCC antibodies (250 ng/ml) for 24 hr, and then measured EPSP amplitude and fixed the cultures for ICC. Representative cultures are shown in (**F**). High magnification images of areas denoted by white squares in left-most panels (**F**) are shown in middle (sensorin immunoreactivity) and right (merged DIC and sensorin, red) panels. Group data show that anti-DCC antibodies significantly decreased sensorin and peIF4E immunoreactivity (**G**) and EPSP amplitude between SN-LFS MNs (**H**). Control cultures were incubated with anti-mouse IgG antibodies (250 ng/ml), which did not affect EPSP amplitude (**H**). Error bars represent SEM. *$p < 0.05$, **$p < 0.01$, unpaired Student's *t*-test. Scale bars in (**A**) and in left panel of (**C**) and (**F**) = 100 μm; in the middle and the right panels of (**C**) and (**F**) =10 μm.

in neurons (*Lebeau et al., 2011*). We monitored translation using quantitative immunocytochemistry for a SN-specific protein, sensorin. We monitored translation more globally by immunocytochemistry for the activated, phosphorylated forms of eIF4E and 4EBP, live imaging with a sensorin-based translational reporter (*Wang et al., 2009*), and the recently described ribopuromylation method to visualize newly synthesized proteins (*David et al., 2012*). The results of our experiments revealed that RNAs and translational machinery are delivered throughout the neuron, but that translation is spatially restricted to sites of synaptic contact or to stimulated synapses. They further indicate that the guidance factor netrin-1 promotes translation at synapses. Our findings are most consistent with a model in which synapse formation leads to the release of netrin-1 from the MN, which in turn triggers pre-synaptic DCC to promote translation in the pre-synaptic compartment.

## Local regulation of translation, rather than RNA targeting, mediates spatial regulation of gene expression during synapse formation

Studies of mRNA localization in non-neuronal asymmetric cells have revealed that a large number of transcripts localize to specific subcellular compartments. For example, high throughput FISH analyses of over 3000 transcripts in *Drosophila* embryos indicated that over 70% of mRNAs were expressed in an array of highly specific subcellular patterns (*Lécuyer et al., 2007*). These findings suggest that targeted localization of mRNA spatially regulates gene expression to essentially define the fate of discrete subcellular compartments. We have previously identified hundreds of mRNAs in the neurites of *Aplysia* SNs (*Moccia et al., 2003*), and studies of localized mRNAs in mammalian hippocampal neurons have indicated that hundreds (*Eberwine et al., 2002*; *Poon et al., 2006*) to thousands of transcripts (*Cajigas et al., 2012*) localize to dendrites. While some studies have indicated that a handful of these mRNAs localize to specific subcellular loci within the neuronal process, including *sensorin* mRNA, which localizes to synapses in Aplysia SNs (*Lyles et al., 2006*), and *arc* mRNA, which localizes to stimulated synapses in mammalian dentate granule neurons (*Steward et al., 1998*), whether or not mRNAs undergo directed targeting from the soma to specific synapses, and whether or not RNA localization defines the fate of specific neuronal compartments, remains an open question.

The *Aplysia* SN-MN culture system provides an ideal preparation for monitoring mRNA localization and local translation during synapse formation and synaptic plasticity, since a single neuron can be manipulated to form synapses or to undergo transcription- and translation-dependent synaptic strengthening in a branch-specific manner. Using this preparation, we find that the local proteome was regulated at the level of translation rather than by stimulus-induced mRNA targeting from the nucleus. Thus, transcripts, ribosomes, and translational machinery were delivered throughout the neuron, but translation was selectively activated in response to local stimuli such as synapse formation. We note that our studies do not rule out a role for more localized subcellular trafficking of mRNAs within neuronal processes, and/or selective stabilization within processes (as described in *Farris et al. (2014)*), but they do indicate that synapse formation and synaptic stimulation do not regulate the trafficking from cell soma to process or synapse. Thus, transcripts that are induced by stimuli are delivered throughout the neuron, with their local distribution and translation being regulated by localized cues.

## Netrin-1 serves as a local cue to spatially regulate translation within neurons

We focused on a role for netrin-1 as the retrograde signal driving translation in the sensory neuron because it has been shown to promote translation in axonal growth cones (*Campbell and Holt, 2001*), to promote synaptogenesis in mammalian cortical neurons (*Goldman et al., 2013*), and because

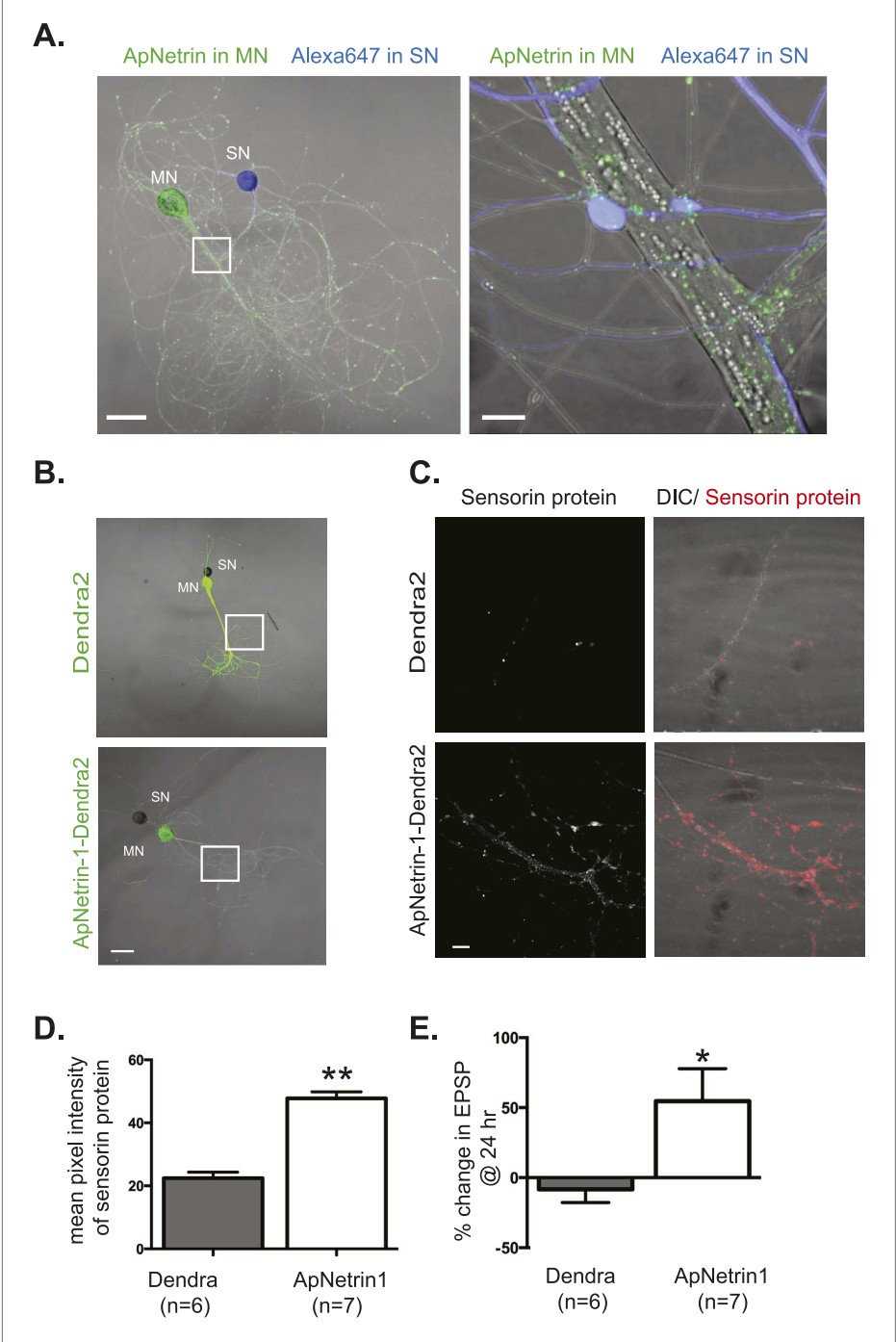

**Figure 9**. Overexpression of *Aplysia* netrin-1 increases translation and synaptic strength in SN-MN synapses. SN-LFS MNs were cultured for 3 days (**A**–**C**). The SN was microinjected with Alexa647 (blue) and the MN with a vector expressing *Aplysia* netrin-1 (ApNetrin-1) c-terminally tagged with dendra2 (green, **A**). The left panel of **A** shows a low magnification image, and the right shows high magnification of area outlined with a white square, as a merged DIC/ApNetrin (green) and Alexa647 (SN, blue) image. ApNetrin-dendra is present in a punctate pattern throughout the MN process (**A**). In (**B**), on DIV3, vectors expressing dendra2 (upper panel) or *Aplysia* netrin-1 c-terminally tagged with dendra2 (lower panel) were microinjected into the LFS MN. Basal EPSPs were measured before and 24 hr after microinjection, and cultures were then processed for immunocytochemistry with anti-sensorin antibodies. Representative high magnification images of areas denoted by white squares in (**B**) are shown in (**C**) with sensorin protein in the left panel and merged image for DIC/sensorin (red) in the right

*Figure 9. Continued on next page*

*Figure 9. Continued*

panel, in cultures in which the MN expresses dendra2 (top) or ApNetrin-dendra2 (bottom). Group data (**D**) show that sensorin immunoreactivity was increased in SNs paired with MNs overexpressing ApNetrin-1. Overexpression of ApNetrin in the MN also increases the EPSP amplitude, while overexpression of dendra2 in the MN has no effect on EPSP amplitude between SN and MN at 24 hr (**E**). Error bars represent SEM. **$p < 0.01$, *$p < 0.05$, unpaired t-test. Scale bars in (**A**) = 100 μm; in (**B**) = 10 μm.

Flanagan et al. have reported that the netrin receptor DCC serves as a transmembrane translation regulation complex in dendrites and axons (*Tcherkezian et al., 2010*). The finding that anti-DCC antibodies decrease sensorin immunoreactivity in SNs paired with MNs, but not in isolated SNs, is consistent with netrin-1 being released from the MN and binding to DCC on the adjacent pre-synaptic SN to promote the translation of sensorin at synapses. Netrin-1 binding to DCC may promote translation by releasing translational machinery from binding DCC cytoplasmic domains within the pre-synaptic terminal, as proposed by Flanagan et al. (2010). Alternatively, netrin-1-induced translation may be downstream of PI3 kinase and/or MAPK pathways, both of which are activated by netrin-1 binding to DCC (*Li et al., 2004*; *Round and Stein, 2007*).

We previously reported that serotonin-induced translation of sensorin in SNs required a calcium-dependent retrograde signal from the MN (*Wang et al., 2009*). In the present study, we show that ApNetrin expressed in MN is released and binds to the SN in a calcium-dependent manner. What the source of calcium is remains an open question. One possibility is that it results from spontaneous release of glutamate from the pre-synaptic SN during synapse formation and/or during synaptic plasticity (*Eliot et al., (1994)*; *Jin et al., (2012)*; *Villareal et al., (2007)*), although see also *Sutton et al. (2006)*, which indicates that spontaneous release serves to suppress local translation in hippocampal neurons.

While we used recombinant human netrin-1 and function-blocking antibodies against human DCC, we are confident that these reagents reflect the endogenous functions of netrin-1 and DCC in *Aplysia.* We cloned *Aplysia* netrin-1 and found that it is 47% identical and 60% similar to human netrin-1 (by comparison, *C. elegans* netrin-1 is 44% identical and 59% similar to human netrin-1). We also cloned *Aplysia* DCC and found that it is 31% identical and 47% similar to human DCC (by comparison, *C. elegans* DCC is 18% identical and 31% similar to human DCC). Moreover, the function-blocking antibody recognizes the extracellular domain, which is the most highly conserved region of the protein (extracellular domain of *Aplysia* DCC is 39% identical and 55% similar to human DCC). Finally, we show that overexpression of ApNetrin in MNs significantly increases synaptic strength and sensorin immunoreactivity (*Figure 9*).

We suspect that netrin-1/DCC signaling is one of multiple local cues and signaling pathways that regulate translation in neurons and that translation is not regulated in a digital on/off manner, but rather in a graded manner. This explains, for example, why there is some sensorin translation in SN neurites contacting non-target L11 (*Figure 5*), and why sensorin protein is expressed in isolated sensory neurons that have been incubated with anti-DCC antibodies (*Figure 8A*).

Our previously published report that synapse formation alters the localization of sensorin mRNA such that it concentrates at synapses (*Lyles et al., 2006*; *Meer et al., 2012*), and that serotonin-regulated translation of a sensorin-based translational reporter was restricted to sites of synaptic contact, suggested that the activity-dependent synaptic localization of mRNA played a critical role in the regulation of its translation. Our current finding that netrin-1 increases translation at synaptic and non-synaptic sites indicates that the precise subcellular localization of the mRNA is secondary to the localization of the netrin-1 signal. Thus, sensorin mRNA does not have to be in the synaptic milieu per se to be translated but rather has to localize to a site in which netrin-1/DCC signaling can occur. We also note that we previously found that there was an increase in the concentration of sensorin mRNA in SN branches contacting target as compared to non-target MNs in more mature (5 DIV) cocultures (*Lyles et al., 2006*). This suggests that while there is not directed targeting of sensorin mRNA from soma to process during synapse formation, there may be preferential stabilization, over time, of sensorin mRNA at sites of synaptic contact. This finding is consistent with recent studies indicating that regulated RNA degradation functions to localize mRNAs and to limit translation within axons (*Colak et al., 2013*).

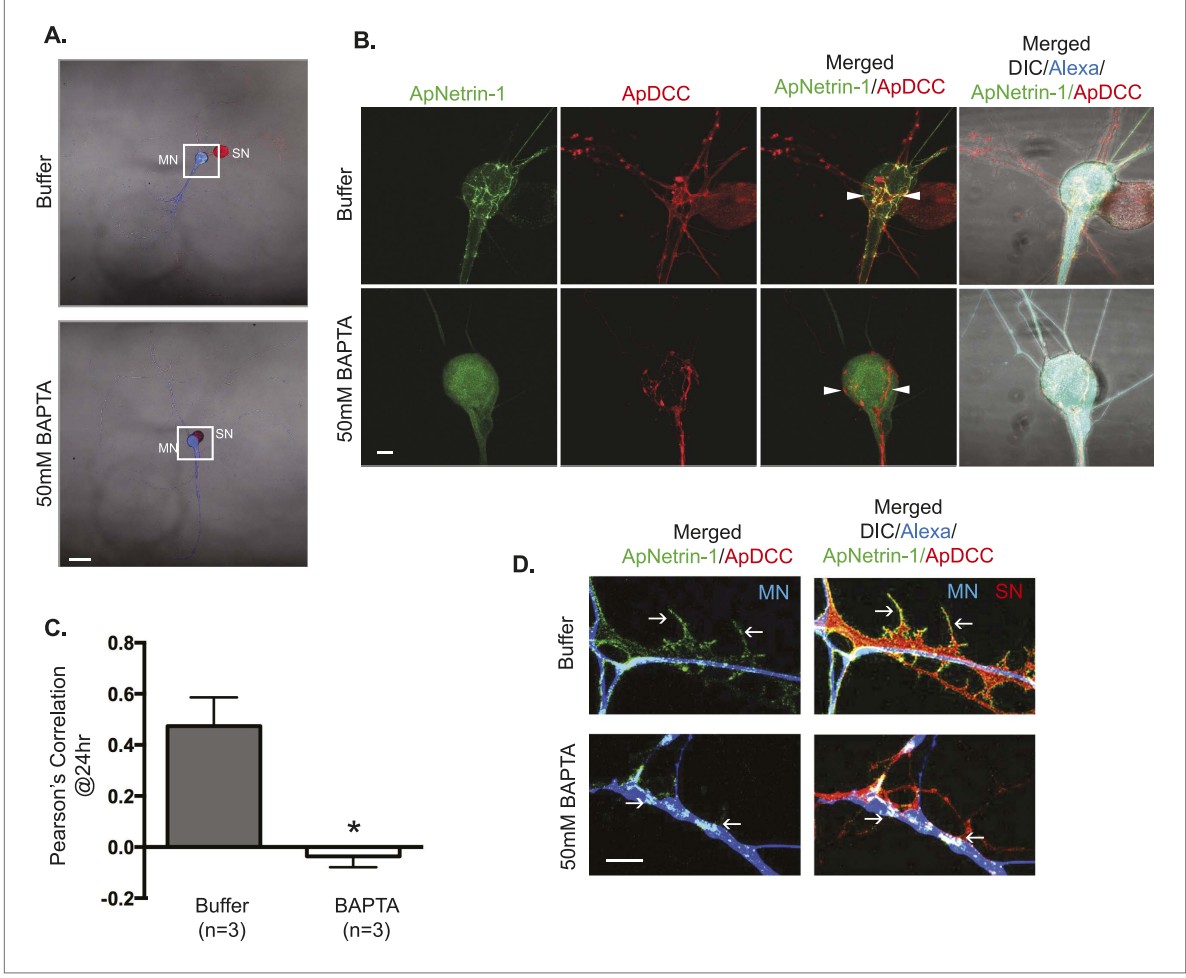

**Figure 10**. ApNetrin-1 undergoes calcium-dependent release from MN and binds SN. SN-LFS MNs were cultured for 2 days, and the SN was microinjected with a vector expressing ApDCC c-terminally tagged with mCherry (red) and the MN with Alexa647 (blue) and with a vector expressing Aplysia netrin-1 (ApNetrin-1) c-terminally tagged with dendra2 (green). In the top panel of (**A**), the MN was microinjected with vehicle; in the bottom panel of (**A**), the MN was microinjected with BAPTA (50 mM) to chelate calcium. Cultures were imaged before and 24 hr after BAPTA or buffer injection later. The representative micrograph in (**A**) shows a low magnification image in which the DIC, ApDCC-mCherry (SN, red), and Alexa647 (MN, blue) signals are merged. (**B**) shows representative images taken 24 hr after microinjection of BAPTA or buffer of the areas highlighted by the white squares in (**A**). The green signal shows the ApNetrin-1 expressed in the MN; the red signal shows ApDCC expressed in the SN, the merged image is of the green ApNetrin-1 (expressed in MN) and red ApDCC (expressed in SN), and the far-right merged image is of DIC, Alexa647 (blue, MN), ApNetrin-1 (green, expressed in MN), and ApDCC (red, expressed in SN). In cultures in which the MN was microinjected with vehicle, the ApNetrin-1 (green) colocalizes with SN processes that are in contact with the MN soma (arrowheads) (top panel, **B**). In cultures in which calcium was chelated in the MN by microinjection of BAPTA, the ApNetrin-1 (green) signal is present within the MN soma, and do not colocalize with the ApDCC (red) in SN processes that are in contact with the MN (arrowheads). The amount of colocalization between the ApNetrin-1 signal and the SN (red) in the soma was quantified by Pearson's correlation (**C**). *p < 0.05, Student's unpaired t-test. Representative images of ApNetrin-1 (green, expressed in MN, blue) signal in distal processes of MNs microinjected with buffer or with BAPTA are shown in (**D**). In control (buffer-injected MNs) cultures, the green ApNetrin-1 signal decorates the red SN processes (arrows). In cultures in which calcium has been chelated by microinjection of BAPTA into the MN, the green ApNetrin-1 signal remains sequestered within the MN (blue). Scale bars in (**A**) =100 μm; in (**B**) = 20 μm; in (**D**) = 10 μm.

The following figure supplement is available for figure 10:

**Figure supplement 1**. Incubation with anti-DCC antibodies inhibits binding of MN-expressed ApNetrin-1 to SNs.

## Local translation contributes substantially to the local proteome

Our study also sheds light on the relative contributions of local translation and somatic translation to the local proteome. Thus, we can compare the amount of netrin-1-induced translation of the sensorin reporter in SN-LFS cocultures with and without the soma (*Figure 7—figure supplement 1D*). Although

one cannot directly compare the total concentration of sensorin in intact SN and isolated SN neurites, our data indicate that netrin-1 is able to induce an equivalent increase in SN neurites in the presence or absence of a SN soma. This suggests that local translation contributes substantially to stimulus-induced changes in the local proteome.

The fact that netrin-1 was able to stimulate global translation at non-synaptic sites as detected by phospho-eIF4E immunocytochemistry (*Figure 7—figure supplement 1C*) suggests that a large number of transcripts, and not just the two mRNAs we detected by FISH, localize throughout the neuronal arbor. Moreover, we found that every type of RNA that we imaged was delivered from the soma throughout the neuronal arbor, whether during synapse formation or following local stimulation. In contrast, we found that every measure of translation that we employed demonstrated localized translation at synapses and/or stimulated synapses. While our results are consistent with the idea that mRNAs are delivered throughout the neuronal arbor, validation of this idea as a general mechanism will require analysis of a larger pool of specific transcripts.

Taken together, our findings show that a signaling pathway involved in axon guidance also regulates synapse formation by spatially restricting neuronal gene expression. Our findings further underscore an uncoupling of transcriptional and translational regulation in neurons, with stimuli triggering transcription of a population of mRNAs that are delivered throughout the neuron, but with local cues independently regulating their translation. Decentralizing the control of gene expression to individual subcellular compartments in this manner enriches the adaptability and plasticity of the nervous system by enabling each compartment to change its proteome in response to local cues. This mechanism allows all the branches of a neuron to be in a state of readiness to respond to local stimuli by changing their local proteome and hence their synaptic structure and function.

## Materials and methods

### *Aplysia* cell culture

Cultures were prepared from adult 80–100 gram *Aplysia* (Alacrity, Redondo Beach, CA) for SN-LFS MN cultures or from the abdominal ganglion of juvenile *Aplysia* (1–4 g, National Aplysia Resource, University of Miami, FL) for L7 and L11 MNs. Animals were anesthetized by injecting 0.35 M $MgCl_2$ into the body, ganglia were dissected and incubated in protease (10 unit/ml protease XIV, Sigma #P5147) in L15 culture media for 1 hr 50 min to digest the connective tissue sheath. Bifurcated SNs from the pedal–pleural ganglia were paired with L7 and L11 MNs from the abdominal ganglion of juvenile *Aplysia* or with two LFS MNs from adult animals. Culture methods are described in detail in *Zhao et al. (2009)*.

To induce LTF of SN-MN synapses, we applied five spaced applications of 5HT (10 μM) using either bath or local perfusion. Bath application of 5x5HT consisted of five 5 min applications of 5HT (10 μM) in L15 media with four intervening 20 min washes in L15, for a total stimulation time of 1 hr 45 min. Local perfusion of 5x5HT consisted of local delivery of 5HT (100 μM) to SN-MN synapses with a perfusion pipette; five applications were given at 10 min intervals, with each application consisting of five 5 s pulses given at 10 s intervals. ASW (Artificial Sea Water) was used as a control for all experiments. Cultures were fixed immediately after the last stimulation or at 24 hr after stimulation and processed for immunocytochemistry or FISH.

### Live cell imaging, microinjection, and electrophysiology

Confocal images were taken on a Zeiss LSM 700 scanning laser microscope. For FISH and ICC experiments, we used a Picospritzer (World Precision Instrument, Sarasota, FL) to microinject 5 mM Alexa647 (bifurcated SN) or Alexa546 (L7 MN) as volume controls and to clearly differentiate SN from MN neurites at least 3 hr before confocal imaging. Image analysis was done by an observer blind to experimental design.

To investigate the distribution of RNA granules during synapse formation, 200 ng/μl ApStaufen-dendra2 (in pNEX3) together with Alexa647 was microinjected into SNs on DIV 1 and imaged on DIV3. For the RNA granules targeting after local perfusion, images were acquired before and 24 hr after local perfusion.

To measure new translation, we microinjected 200 ng/μl sensorin-dendra2 reporter (in pNEX3, described in *Wang et al. (2009)*) into SNs on DIV 1. On day 2, we removed the SN cell body and 18 hr later, we photoconverted dendra2 from green to red as a described in *Wang et al. (2009)*, with slight modifications, using an X-cite series120Q excitation light (a 120-watt lamp), delivering two 5 s

pulses of UV illumination with a 10 s interval. After imaging, we returned cultures to the 18°C incubator and imaged the green and red signals 24 hr later. To compare new translation in SN branches contacting target or non-target motor neuron, we measured total fluorescence (area (μm2) × mean fluorescence intensity) of both green and red fluorescence using Slidebook5 software. New translation was normalized as the total pixel intensity in the dendra2 green signal to the dendra2 red signal 24 hr after photoconversion.

To test whether ApNet-1 depends on post-synaptic $Ca^{2+}$, 200 ng/μl dendra2-tagged ApNet-1 (green) was microinjected to the LFS MN and 200 ng/μl ApDCC-mCherry (red) was microinjected into the SN at DIV2. 24 hrs later, 50 mM BAPTA in Buffer (1.5 M K-Acetate, 0.5 M KCl, 0.01 M HEPES, pH7.2) was microinjected into the post-synaptic LFS MN. Buffer alone was microinjected as a control. Images were taken pre- and 24 hr after BAPTA or buffer microinjection.

To test the effect of anti-DCC antibodies on the binding of MN-expressed ApNetrin-1-dendra2 to ApDCC-mCherry-expressing SNs, cultures were incubated with anti-DCC antibodies (250 ng/ml) for 24 hr. In control cultures, ASW or non-immune IgG (250 ng/ml) was added for 24 hr. Images were taken 24 hr after incubation.

To measure varicosity number in SN-MN cocultures, a VAMP (synaptobrevin)-mCherry pNEX3 construct was expressed in SNs paired with LFS MNs for 48 hr. The number of varicosities was measured before and 24 hr after incubation with 250 ng/ml netrin-1.

EPSP amplitude was measured as described in *Zhao et al. (2003)*. Briefly, LFS MNs were impaled with a sharp glass electrode (10–15 MΩ) filled with 2 M potassium acetate, and the membrane potential of MNs was held at −80 mV and of SNs at −50 mV. EPSP amplitudes were measured before and 24 hr after 250 ng/ml netrin-1 treatment in L15 media using Axoscope 8.2 and pCLAMP 8 (Axon Instruments, Union City, CA) following intracellular depolarization (1–3 nA for 5 ms) of the SN. Artificial Sea Water (ASW) was used as a vehicle control.

## Ribopuromycylation

We slightly modified the ribopuromycylation method described in *David et al. (2012)*. Specifically, we cultured Aplysia SNs with L7 target MNs and L11 non-target MNs for 3DIV and then replaced the culture media with L15 media. Cultured neurons were incubated with 200 μM emetine in L15 media for 30 min before adding puromycin. After 30 min, 100 μM puromycin with 200 μM emetin in L15 media was added for 10 min. Cultured neurons were then washed with L15 and immediately fixed with 4% PFA/30% sucrose for 30 min.

## Fluorescence in situ hybridization (FISH) and immunocytochemistry (ICC)

FISH was performed as described in *Lyles et al. (2006)* with slight modifications. Briefly, cultured neurons were fixed with 4% PFA/30% sucrose for 30 min at room temperature. For FISH, after fixation, we hybridized with 40 ng/ml 18S rRNA, 28S rRNA, EF1α RNA, or sensorin RNA probed and imaged after FISH. For β-thymosin RNA FISH, we followed by the protocol as described in ViewRNA ISH processes (Affymetrix, Santa Clara, CA). Image settings were adjusted so that sense 18S rRNA, sense 28S rRNA, sense EF1α RNA, and sense sensorin mRNA probes did not produce any signal and no probes were used as negative control for β-thymosin RNA.

To generate riboprobes, we performed in vitro transcription with T7/T3 polymerase, tagging riboprobes with digoxigenin or biotin and purifying with ProbeQuant G-50 microcolumn (#28-9034-08; GE healthcare, Piscataway, NJ). Before we used riboprobes, we measured their concentration and tested their specificity by dot blotting with anti-digoxigenin-POD (#11207733910; Roche, Indianapolis, IN) or HRP-mouse anti-biotin (#03-3720; Invitrogen, Grand Island, NY). All FISH experiments were performed with sense riboprobes as a control to measure background signals. FISH detection using the TSA amplification system was followed as a described in *Lyles et al. (2006)*.

To investigate netrin-1 mediated translation, 250 ng/ml recombinant Fc human netrin-1 (#ALX-522-455; Enzo life sciences, Farmingdale, NY) in L15 media or functional blocking anti-DCC (#OP45; Milipore, Billerica, MA) was applied to cultures for 24 hr and then washed with L15 media. Cultures were immediately fixed with 4% PFA/30% sucrose for 30 min.

After fixation, ICC was performed as described in *Martin et al. (1997)*. Primary antibodies included custom-made chicken anti-sensorin antibodies (generated against peptide CATRSKNNVPRRFPRARY RVGYMF by Aves Labs, Inc., Tigard, OR), rabbit anti-eIF4E (#9742; Cell Signaling, Beverly, MA), rabbit

anti-phospho eIF4E (#9741; Cell Signaling), rabbit anti-4EBP (#9452; Cell Signaling), rabbit anti-phospho 4EBP (#2855; Cell Signaling), and mouse monoclonal anti-puromycin, clone 12D100 (Millipore, #MABE343).

## Cloning of *Aplysia* netrin and DCC

Degenerate primers were used to amplify segments of netrin-1 and DCC from RNA isolated from adult *Aplysia* CNS. The sequenced fragments were BLAST searched against the following database: http://aplysiagenetools.org/. Identified hits were used to generate primers that were in turn used to amplify full-length *Aplysia* netrin-1 (GenBank accession #KM218335) and DCC (accession #KM218336) from RNA isolated from CNS. Full-length clones were verified by sequencing.

## Statistical analysis

All results are presented as mean ± SEM. Prism GraphPad was used to perform all statistical analysis. The type of statistical analysis performed is described in each figure legend.

## Acknowledgements

We thank S Haasan for blind image analysis, D Black, JT Braslow, P Chen, TH Ch'ng, VM Ho, GM Martin, K Olofsdotter-Otis, B Uzgil, and L Zipursky for comments on the manuscript, and all members of the Martin lab for helpful discussions. The work was supported by NIH R01NS045324 (to KCM).

## Additional information

### Funding

| Funder | Grant reference number | Author |
| --- | --- | --- |
| National Institutes of Health | R01NS045324 | Kelsey C Martin |

The funder had no role in study design, data collection and interpretation, or the decision to submit the work for publication.

### Author contributions

SK, KCM, Conception and design, Acquisition of data, Analysis and interpretation of data, Drafting or revising the article

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
