## [Decision Letter]

Thank you for sending your work entitled “Neuron-wide RNA transport combines with netrin-mediated local translation to spatially regulate the synaptic proteome” for consideration at *eLife*. Your article has been quite favorably evaluated by a Senior editor and 3 reviewers, one of whom, Mani Ramaswami, is a member of our Board of Reviewing Editors. Kang Shen, another of the peer reviewers, has also agreed to reveal his identity.

The Reviewing editor and the other reviewers discussed their comments before we reached this decision, and the Reviewing editor has assembled the following comments to help you prepare a revised submission.

Kim and Martin describe the transport of RNAs in a cultured sensory neuron that forms connection with two motor neurons: a synaptic connection with L7 and a non-synaptic adhesive contact with L11. The authors show fairly convincingly that, in these preparations, three different RNA species which are: (a) constitutively expressed; (b) trancriptionally induced by activity, or; (c) translationally controlled by activity are distributed equally between the two sensory neuron axons/ neurite. However, the last class (sensorin) is translated more efficiently in the synaptically connected branch. This increased translation correlates with increased eIF4e and 4-EBP phosphorylation. They further show that externally applied netrin, suggested to work as a translational activator and a synaptic signaling molecule can induce translation of sensorin RNA in SNs through a pathway that requires endogenous DCC and involves eIF4e phosphorylation. They also show that forced netrin expression in MNs induced sensorin translation in presynaptic SNs and that this requires calcium signaling in MNs.

Despite the general clarity of the paper, there are several significant concerns.

Do note that the major comment 1 must be addressed in the revised manuscript. Comments 2 and 3 also require additional experiments, which the authors must seriously consider as they would greatly strengthen the conclusions made in the paper.

Major points:

1) The paper is almost entirely dependent on a preparation in which spontaneous synaptic activity is (reasonably) inferred in one neurite and believed to induce RNA transport granule disassembly and RNA translation in that neurite. Thus, it describes a developmental process that may or may not involve mechanisms involved in learning and memory. The manuscript does not make this distinction clear.

The authors must perform the same experiments with pulsed 5HT stimulation to look at neurotransmitter induced protein synthesis and granule disassembly. This would serve to integrate their observations with an established *Aplysia* literature and link it to previous studies (substantially from the authors' lab) of serotonin-induced translation and synaptic plasticity. In addition, it would allow the authors to make a comparison, in the same preparation, between mechanisms of translational control (including mRNA and RNA granule dynamics) involved in 5HT-induced translation and those associated with spontaneous-activity regulated translation.

The revised paper should be rewritten to clearly discriminate between translational control events induced by acute 5HT stimulation and those associated with spontaneous activity.

With respect to the latter, the experiments were performed in young neurons (day 3), did the author analyse additional time points? The outcome might be different according to the developmental stages. This aspect should be discussed.

2) There is no evidence that netrin is expressed in MNs and that it is normally required for sensorin translation in SN neurites. The experiments presented, including the requirement for DCC, only show that netrin has this potential ability, not that this is its biological role. Can this issue be experimentally addressed? Does netrin have an effect on sensorin translation in MNs as well? In Figure 10, one potential control experiment for this is to use the DCC antibody to see if the labeling of SN axon by the labeled netrin is based on netrin-DCC binding.

3) Does *Aplysia* Staufen bind sensorin and EF1alpha mRNA? The authors should also ask if endogenous Staufen behaves similarly to the overexpressed form (ApStaufen-dendra2). These are important points because overexpression of RNA binding proteins very often leads to aggregates formation. Caveats from this complication should be clearly addressed in the text and hopefully through experiments.

4) Can the authors examine additional mRNAs to see if they are also globally transported in response to synaptic activity?

5) Six figures to characterize RNA transport and translational in this *Aplysia* preparation seem too many. I think the key data could be safely and better compressed into three figures.

---

## [Author Response]

*Do note that the major comment 1 must be addressed in the revised manuscript. Comments 2 and 3 also require additional experiments, which the authors must seriously consider as they would greatly strengthen the conclusions made in the paper*.

We have performed three additional experiments in response to the reviewers’ comments, which we summarize below.

1) We analyzed the distribution of another localized mRNA in *Aplysia* sensory neurons, encoding β-thymosin. As shown in Figure 2—figure supplement 2, like sensorin mRNA, β-thymosin mRNA also distributes equally well to sensory processes contacting target and nontarget motor neurons. These data support the generality of our finding that mRNAs distribute throughout the neuronal arbor rather than being delivered from the nucleus and soma to a specific synaptic target.

2) We examined the distribution of ApStaufen-dendra2 in bifurcated sensory neurons following local stimulation with 5x5HT. These data, shown in Figure 4–figure supplement, indicate that local stimulation induces an increase in the number of Staufen-containing puncta in the sensory neuron processes, but that there is no difference in the increase between the stimulated and unstimulated branch. Again, these data are consistent with RNAs being delivered throughout the neuronal process following local stimulation.

3) We incubated cultures of motor neurons expressing ApNetrin-Dendra2 and sensory neurons expressing DCC-mCherry with anti-DCC antibodies (or with IgG as a negative control). These experiments, shown in Figure 10—figure supplement 1, show that the anti-DCC antibodies greatly reduce the binding of ApNetrin-1-dendra2 to the sensory neurons. This control supports our conclusion that ApNetrin-1 is released from motor neurons and binds to DCC on the sensory neuron.

*Major points*:

*1) The paper is almost entirely dependent on a preparation in which spontaneous synaptic activity is (reasonably) inferred in one neurite and believed to induce RNA transport granule disassembly and RNA translation in that neurite. Thus, it describes a developmental process that may or may not involve mechanisms involved in learning and memory. The manuscript does not make this distinction clear*.

While the *Aplysia* sensory-motor culture system is best known in the context of learning and memory, it also presents many advantages to the study of synapse formation since it is, to our knowledge, the only preparation in which one can culture neurons that 1) do not form chemical autapses and 2) that will fasciculate with but not form chemical synapses with a non-target follower neuron. Several published studies have taken advantage of this phenomenon (e.g. [14]; [22]; [32]; [35]) though, as the reviewer points out, certainly not as many as have focused on learning-related synaptic plasticity. We very explicitly set out to study RNA localization and translation in the context of both local synapse formation and local 5HT-induced synaptic plasticity in our study. In particular, our aim was to examine a single neuron with two branches that received localized signals either as a result of synapse formation or as a result of local 5HT stimulation, since we recognized that this allowed us to study more broadly whether and how localized signals received at localized and distal subcellular compartments could regulate the trafficking of mRNAs out of the soma. We understand that the use of both types of preparations may generate some confusion and have done our best to revise the text to more clearly differentiate between the experiments involving synapse formation and those involving plasticity induced by acute 5HT stimulation.

We would also like to respond to the reviewer's comment here and elsewhere about granule disassembly by pointing out that our methods do not allow us to study granule assembly or disassembly. Rather, our focus is on RNA distribution within the neuron, and on sites of translation. While the biology of RNA granule assembly and disassembly is important and highly relevant to our studies, it is not the focus of our experiments.

*The authors must perform the same experiments with pulsed 5HT stimulation to look at neurotransmitter induced protein synthesis and granule disassembly. This would serve to integrate their observations with an established* Aplysia *literature and link it to previous studies (substantially from the authors' lab) of serotonin-induced translation and synaptic plasticity. In addition, it would allow the authors to make a comparison, in the same preparation, between mechanisms of translational control (including mRNA and RNA granule dynamics) involved in 5HT-induced translation and those associated with spontaneous-activity regulated translation.*

In response to the reviewer’s request that we perform additional experiments with pulsed 5HT stimulation, we have now added an experiment in Figure 4–figure supplement 4 in which we locally perfuse a bifurcated sensory neuron with 5x5HT and examine the localization of ApStaufen-dendra2. We have previously published studies of local translation following local stimulation with 5x5HT (48) and that data matches the data on local translation during synapse formation shown in Figure 5. We refer to these results in the Results section and in the Discussion section.

*The revised paper should be rewritten to clearly discriminate between translational control events induced by acute 5HT stimulation and those associated with spontaneous activity*.

*With respect to the latter, the experiments were performed in young neurons (day 3), did the author analyse additional time points? The outcome might be different according to the developmental stages. This aspect should be discussed*.

In response to the question of whether we have analyzed mRNA localization at other time points, we include data examining sensorin mRNA localization in DIV1 neurons in Figure 2—figure supplement 1. We have previously published data examining sensorin mRNA localization in DIV5 neurons, and include a discussion of this data in the Discussion section.

*2) There is no evidence that netrin is expressed in MNs and that it is normally required for sensorin translation in SN neurites. The experiments presented, including the requirement for DCC, only show that netrin has this potential ability, not that this is its biological role. Can this issue be experimentally addressed? Does netrin have an effect on sensorin translation in MNs as well? In*
Figure 10*, one potential control experiment for this is to use the DCC antibody to see if the labeling of SN axon by the labeled netrin is based on netrin-DCC binding*.

The reviewer raises an important point concerning the necessity of netrin-1 in promoting sensorin translation. Unfortunately, we have had technical difficulties in assaying netrin-1 expression in single SNs or MNs by FISH. We believe that this is because the RNAs are likely present at low concentrations, since when we overexpress ApNetrin-1, we can detect specific signal by FISH. Consistent with the idea that endogenous ApNetrin-1 (or another DCC ligand) does function to promote translation in sensory neurons, we show that function-blocking DCC antibodies prevent the induction of sensorin translation that is normally observed in SN upon pairing with a MN (Figure 8). As far as any effect of ApNetrin-1 on sensorin translation in MNs, we note that the MN does not express sensorin mRNA. Finally, we have now performed the experiment suggested by the reviewer, in which we use the DCC antibody to see if the labeling of the SN axon by the labeled netrin is based on netrin-DCC binding. This data is shown in Figure 10—figure supplement 1. Incubation of cultures with anti-DCC antibodies greatly inhibits the labeling of the SN with the ApNetrin expressed in the MN, consistent with ApNetrin binding to DCC on the SN.

*3) Does* Aplysia *Staufen bind sensorin and EF1alpha mRNA? The authors should also ask if endogenous Staufen behaves similarly to the overexpressed form (ApStaufen-dendra2). These are important points because overexpression of RNA binding proteins very often leads to aggregates formation. Caveats from this complication should be clearly addressed in the text and hopefully through experiments.*

We do not know the identity of the mRNAs bound by *Aplysia* Staufen (ApStaufen). Unfortunately, we do not have an antibody that recognizes ApStaufen by immunocytochemistry or immunoprecipitation, and as a result cannot use immunocytochemistry to visualize endogenous ApStaufen or to immunoprecipitate ApStaufen from *Aplysia* CNS to characterize the bound RNAs (which is a tricky experiment even with a good antibody). Further, since we are expressing tagged ApStaufen by microinjection of individual neurons, we cannot perform any biochemical assays to identify the bound mRNAs (since there is not enough starting material). While we hope to generate anti-ApStaufen antibodies in the future that will allow us to perform these experiments, we also think these types of experiments are better done, as they have been, in the mouse, *C. elegans* and *Drosophila* systems.

*4) Can the authors examine additional mRNAs to see if they are also globally transported in response to synaptic activity*?

In response to the request to study other localized RNAs, we now include an analysis of the mRNA encoding β-thymosin. As shown in Figure 2—figure supplement 2 of the revised manuscript, these experiments show that, like sensorin and EF1α mRNAs, β-thymosin mRNA localizes equally well to SN branches contacting target and nontarget MNs.

*5) Six figures to characterize RNA transport and translational in this* Aplysia *preparation seem too many. I think the key data could be safely and better compressed into three figures.*

We respectfully disagree with the suggestion that we merge the first six figures into three. Each figure makes a distinct point. The first figure focuses on rRNA distribution. The second figure focuses on the localization of constitutively expressed mRNAs during synapse formation. The third figure focuses on the localization of stimulus-induced mRNAs following branch-specific plasticity. The fourth figure focuses on the Staufen RNA binding protein. We believe that it is important to discriminate between these various classes of RNAs by dividing them into distinct figures (each of which is quite complicated on its own), and are concerned that merging them into one might generate more confusion. The fifth figure shows translation, assayed by immunocytochemistry using sensorin and translation factor antibodies, and by using a live reporter to demonstrate that the translation is local. The sixth figure uses ribopuromycylation to show that this local restriction of translation likely involves a large number of mRNAs, which indicates that many RNAs are distributed equally well to SN branches contacting target and nontarget MNs. We could move this to a supplementary figure if the editors and reviewers felt this was important, but again we believe that the data in each figure make important and distinct points.